# PROSAMPLER: IMPROVING CONTRASTIVE LEARNING BY BETTER MINI-BATCH SAMPLING

## ABSTRACT

In-batch contrastive learning has emerged as a state-of-the-art self-supervised learning solution, with the philosophy of bringing semantically similar instances closer while pushing dissimilar instances apart within a mini-batch. However, the in-batch negative sharing strategy is limited by the batch size and falls short of prioritizing the informative negatives (i.e., hard negatives) globally. In this paper, we propose to sample mini-batches with hard negatives on a proximity graph in which the instances (nodes) are connected according to the similarity measurement. Sampling on the proximity graph can better exploit the hard negatives globally by bridging in similar instances from the entire dataset. The proposed method can flexibly explore the negatives by modulating two parameters, and we show that such flexibility is the key to better exploit hard negatives globally. We evaluate the proposed method on three representative contrastive learning algorithms, each of which corresponds to one modality: image, text, and graph. Besides, we also apply it to the variants of the InfoNCE objective to verify its generality. Results show that our method can consistently boost the performance of contrastive methods, with a relative improvement of 2.5% for SimCLR on ImageNet-100, 1.4% for SimCSE on the standard STS task, and 1.2% for GraphCL on the COLLAB dataset.

## 1 INTRODUCTION

Contrastive learning has been the dominant approach in current self-supervised representation learning, which is applied in many areas, such as MoCo (Chen et al., 2020) and SimCLR (He et al., 2020) in computer vision, GCC (Qiu et al., 2020) and GraphCL (You et al., 2020) in graph representation learning and SimCSE (Gao et al., 2021) in natural language processing. The basic idea is to decrease the distance between the embeddings of the same instances (positive pair) while increase that of the other instances (negative pair). These important works in contrastive learning generally follow or slightly modify the framework of *in-batch contrastive learning* as follows:

$$\text{minimize } \mathbb{E}_{\{x_1 \dots x_B\} \subset \mathcal{D}} \left[ -\sum_{i=1}^{B} \log \frac{e^{f(x_i)^T f(x_i^+)}}{e^{f(x_i)^T f(x_i^+)} + \sum_{j \neq i} e^{f(x_i)^T f(x_j)}} \right], \tag{1}$$

where $\{x_1 \dots x_B\}$ is a mini-batch of samples (usually) sequentially loaded from the dataset $\mathcal{D}$, and $x_i^+$ is an augmented version of $x_i$. The encoder $f(\cdot)$ learns to discriminate instances by mapping different data-augmentation versions of the same instance (positive pair) to similar embeddings, and mapping different instances in the mini-batch (negative pair) to dissimilar embeddings. The key to efficiency is *in-batch negative sharing strategy* that any instance within the mini-batch is the other instances' negative, which means we learn to discriminate all $B(B-1)$ pairs of instances in a mini-batch while encoding each instance only once. Its advantages of simplicity and efficiency make it more popular than pairwise (Mikolov et al., 2013) or triple-based methods (Schroff et al., 2015; Harwood et al., 2017), gradually becoming the dominating framework for contrastive learning.

However, the performance of in-batch contrastive learning is closely related to *batch size*. It is the target of many important contrastive learning methods to obtain a large (equivalent) batch size under limited computation and memory budget. For example, Memory Bank (Wu et al., 2018) stores the encoded embeddings from previous mini-batches as extra negative samples, and MoCo (He et al., 2020) improves the consistency of the stored negative samples via a momentum encoder. SimCLR

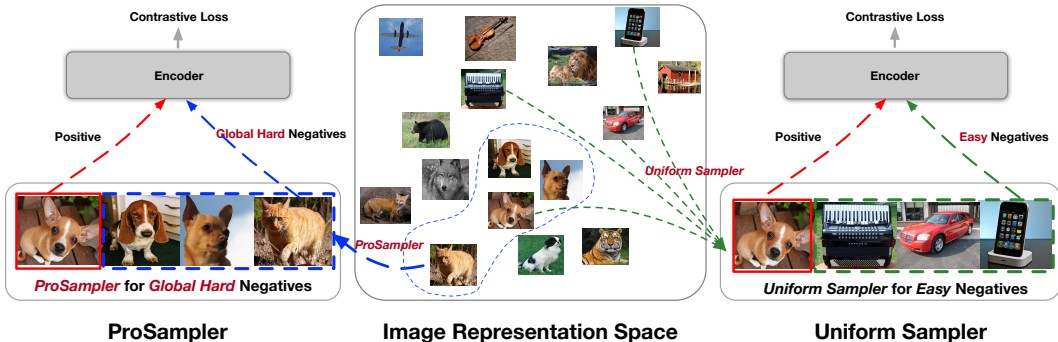

Figure 1: A motivating example of ProSampler. The generated image representations form an embedding space where Uniform Sampler randomly samples a mini-batch with easy negatives and ProSampler samples a mini-batch with hard negatives based on proximity graph.

shows that simply increasing the batch size of the plain in-batch contrastive learning to 8,192 outperforms previous carefully designed methods. Although many works highlight the importance of batch size, a further question arises — which instances in the mini-batch contribute the most?

**Hard negative pair contributes the most** — a clear answer to this question is well-supported by many efforts in some related studies on negative sampling (Ying et al., 2018; Yang et al., 2020; Huang et al., 2021; Kalantidis et al., 2020; Robinson et al., 2021). An intuitive explanation is that the $e^{f(x_i)^T f(x_j)}$ for easy-to-discriminate negative pairs will become very small after the early period of training, and thus the hard negative pairs contribute the majority of the loss and gradients.

The hard negative sampling already made great success in many real-world applications, e.g., 8% improvement of Facebook search recall (Huang et al., 2020) and 15% relative gains of Microsoft retrieval engine (Xiong et al., 2020). The key to these methods is to *globally* select negatives that are similar to the query one across the whole dataset. However, previous methods for negative sampling of in-batch contrastive learning (Robinson et al., 2021; Chuang et al., 2020) focus on identifying negative samples within the current mini-batch, which is insufficient to mine the meaningful negatives from the entire dataset. Meanwhile, previous global negative samplers apply triplet loss and explore the negatives in pairs (Karpukhin et al., 2020; Xiong et al., 2020), which is inapplicable to in-batch negative sharing strategy, since it cannot guarantee the similarity between every instance pair within a mini-batch.

In this paper, we focus on designing a global hard negative sampler for in-batch contrastive learning. Since every instance serves as the negative to the other instances in the same batch, the desired sampling strategy should be the one with more hard-to-distinguish pairs in each sampled batch. This objective can be considered as sampling a batch of similar instances from the dataset. But how can we identify such batch globally over the dataset?

**Present Work.** Here we propose Proximity Graph-based Sampler (ProSampler), a global hard negative sampling strategy that can be plugged into any in-batch contrastive learning method. Proximity graph breaks the independence between different instances and captures the relevance among instances to better perform global negative sampling. As shown in Figure 1, similar instances form a local neighborhood in the proximity graph where ProSampler performs negative sampling as short random walks to effectively draw hard negative pairs. Besides, ProSampler can flexibly control the hardness of the sampled mini-batch by modulating two parameters. In practice, we build the proximity graph per fixed iterations, and then apply Random Walk with Restart (RWR) per iteration to sample a mini-batch for training.

Our experiments show that ProSampler consistently improves top-performing contrastive learning algorithms in different domains, including SimCLR (Chen et al., 2020) and MoCo v3 (Chen et al., 2021) in CV, SimCSE (Gao et al., 2021) in NLP, and GraphCL (You et al., 2020) in graph learning by merely changing the mini-batch sampling step. To the best of our knowledge, ProSampler is the first algorithm to optimize the mini-batch sampling step for better negative sampling in the current in-batch contrastive learning framework.

## 2 RELATED WORK

**Contrastive learning in different modalities.** Contrastive learning follows a similar paradigm that contrasts similar and dissimilar observations based on noise contrastive estimation (NCE) (Gutmann and Hyvärinen, 2010; Oord et al., 2018). The primary distinction between contrastive methods of different modalities is how they augment the data. As for computer vision, MoCo (He et al., 2020), SimCLR (Chen et al., 2020), SwAV (Caron et al., 2020), and BYOL (Grill et al., 2020) augment data with geometric transformation and appearance transformation. Besides simply using data augmentation, WCL (Zheng et al., 2021) additionally utilizes an affinity graph to construct positive pairs for each example within the mini-batch. As for language, CLEAR (Wu et al., 2020b) and COCO-LM (Meng et al., 2021) augment the text data through word deletion, reordering, and substitution, while SimCSE (Gao et al., 2021) obtains the augmented instances by applying the standard dropout twice. As for graph, DGI (Petar et al., 2018) and InfoGraph (Sun et al., 2019) treat the node representations and corresponding graph representations as positive pairs. Besides, GCC (Qiu et al., 2020) and GraphCL (You et al., 2020) augment the graph data by graph sampling or proximity-oriented methods. Zhu et al. (2021) compares different kinds of graph augmentation strategy. Our proposed ProSampler is a general mini-batch sampler which can directly be applied to any in-batch contrastive learning framework with different modalities.

**Negative sampling in contrastive learning.** Previous studies about negative sampling in contrastive learning roughly fall into two categories: **(1) Memory-based negative sampling strategy**, such as MoCo (He et al., 2020), maintains a fixed-size memory bank to store negatives which are updated regularly during the training process. MoCHI (Kalantidis et al., 2020) proposes to mix the hard negative candidates at the feature level to generate more challenging negative pairs. MoCoRing (Wu et al., 2020a) samples hard negatives from a defined conditional distribution which keeps a lower bound on the mutual information. **(2) In-batch negative sharing strategy**, such as SimCLR (Chen et al., 2020) and MoCo v3 (Chen et al., 2021), adopts different instances in the current mini-batch as negatives. To mitigate the false negative issue, DCL (Chuang et al., 2020) modifies the original InfoNCE objective to reweight the contrastive loss. Huynh et al. (2022) identifies the false negatives within a mini-batch by comparing the similarity between negatives and the anchor image's multiple support views. Additionally, HCL (Robinson et al., 2021) revises the original InfoNCE objective by assigning higher weights for hard negatives among the mini-batch. However, such locally sampled hard negatives cannot exploit the hard negatives sufficiently from the dataset.

Global hard negative sampling methods on triplet loss have been widely investigated, which aim to globally sample hard negatives for a given positive pair. For example, Wang et al. (2021) proposes to take rank-k hard negatives from some randomly sampled negatives. Xiong et al. (2020) globally samples hard negatives by an asynchronously-updated approximate nearest neighbor (ANN) index for dense text retrieval. Different from the abovementioned methods which are applied to a triplet loss for a given pair, our ProSampler samples mini-batches with hard negatives for InfoNCE loss.

**Self-supervised learning without negative sampling.** Recently, some attempts on learning without negative sampling achieve promising results, such as BYOL (Grill et al., 2020), SwAV (Caron et al., 2020), SimSiam (Chen and He, 2021), and DINO (Caron et al., 2021). These methods apply Siamese network structure, and contrast the output of an online network and a target network with different augmented views. The main difference between them is how they prevent model from collapsing.

## 3 LEARNING WITH PROSAMPLER

### 3.1 GLOBAL HARD NEGATIVE SAMPLING FOR IN-BATCH CONTRASTIVE LEARNING

Contrastive learning aims to learn a proper transformation that maps two semantically similar instances $x_i, x_j$ to two close points in the embedding space. It applies NCE objective (Gutmann and Hyvärinen, 2010; Oord et al., 2018) and in-batch negative sharing strategy to boost the training efficiency, which means that every instance serves as a negative to the other instances within the mini-batch. How to sample a mini-batch with hard negatives for contrastive learning remains an open problem, and previous methods achieve this by sampling within the mini-batch (Chuang et al., 2020; Robinson et al., 2021; Karpukhin et al., 2020). However, the batch size is far smaller than the dataset

size and sampling within the mini-batch cannot effectively explore the hard negatives from the whole dataset (Xiong et al., 2020; Zhang et al., 2013).

In this work, we delve deeper into learning with the global hard negative sampler, which picks a batch of instances containing considerable hard-to-distinguish pairs. Besides, a desired mini-batch sampling strategy should be general and adaptable to the datasets with various modalities and different scales. We formulate the problem as:

**Problem 1.** *Given a set of data instances $\mathcal{D} = \{x_1, \cdots, x_N\}$, our goal is to design a modality-independent sampler $g(D) = \{x_i, \cdots, x_{(i+B)}\}$ to sample a mini-batch of instances where any instance pair are hard to distinguish across the dataset.*

### 3.2 TWO EXTREME STRATEGIES: UNIFORM SAMPLER AND kNN SAMPLER

Here we discuss two extreme mini-batch sampling strategies for in-batch contrastive learning: Uniform Sampler and kNN Sampler, which represent extreme scenarios in terms of the hardness of a mini-batch they construct.

**Uniform Sampler** is the most common strategy used in contrastive learning (Chen et al., 2020; Gao et al., 2021; You et al., 2020), which is general, easy to implement, and model-independent. The overall pipeline is to first randomly sample a batch of instances for each training step, then feed them into the objective function.

**kNN Sampler** can globally sample a mini-batch with many hard negative examples. As its name indicates, kNN Sampler would pick an instance at random and retrieve a set of nearest neighbors to construct a batch. Figure 4 shows that the mini-batch sampled by kNN Sampler has a high percentage of similar instance pairs.

However, the abovementioned two methods suffer from the following limitations:

- Uniform Sampler neglects the effect of hard negatives (Kalantidis et al., 2020; Robinson et al., 2021), and will select negatives with low gradients that contribute little to optimization. As shown in Figure 4, Uniform Sampler results in a low percentage of similar instance pairs in a mini-batch. Yang et al. (2020) and Xiong et al. (2020) also theoretically prove that the suggested sampled negative should be similar to the query instance since it can provide a meaningful gradient to the model.

- During the self-supervised training, the instances of the same class will cluster together in the embedding space (Chen et al., 2020; Caron et al., 2020). Hence the kNN Sampler can first retrieve the hard negatives but they will be replaced by false negatives (FN) as the training epochs increase. Figure 4 also demonstrates that kNN Sampler exhibits a very high percentage of FN in a mini-batch.

In conclusion, Uniform Sampler cannot leverage hard negatives to guide the optimization of the model; whereas kNN Sampler explicitly samples hard negatives but suffers from the false negative issue. Both of them will result in sub-optimal performance. A better global hard negative sampler for in-batch contrastive learning should trade-off these two sampling styles, and balance the exploitation of hard negatives and the FN issue. Building on the above observations, we propose ProSampler, a flexible global mini-batch sampler which allows us to smoothly interpolate between the kNN Sampler and the Uniform Sampler.

### 3.3 PROSAMPLER

As discussed in Section 3.1, a desired mini-batch should be the one where any example is the hard negative of the other examples. This objective can also be seen as sampling a group of instances which are close to one another in the embedding space. But how to identify such groups globally from the dataset? As shown in Figure 2, we propose to capture similarity relationships among instances by *proximity graph*. Proximity graph connects the instances by the similarity measurement, and in this way, instances that appear to be close to each other form a local community in the graph. We perform the mini-batch sampling as a walk in the proximity graph, which collects the visited instances as sampling results. To modulate the hardness of a sampled batch, we introduce two parameters $M$ and $\alpha$ to control the behaviors of proximity graph construction and sampling respectively.

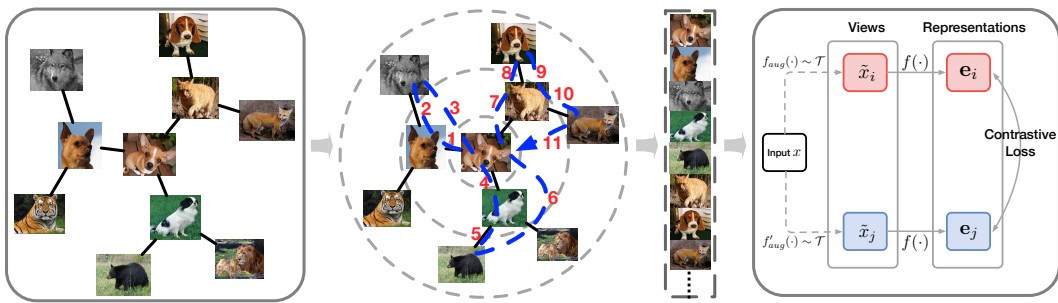

**Proximity Graph Construction**  **Proximity Graph Sampling**  **Batch**  **In-batch Contrastive Learning**

Figure 2: The framework of ProSampler. The proximity graph is first constructed based on generated image representations and will be updated every $t$ training steps. Next, a proximity graph-based negative sampler is applied to generate a batch with hard negatives for in-batch contrastive learning.

**Proximity graph construction.** Recall that in a training dataset, we have $N$ observations $\{v_i | i = 1, \cdots, N\}$ and their corresponding representations $\{\mathbf{e}_i | i = 1, \cdots, N\}$ generated by the current encoder $f(\cdot)$. We formulate the proximity graph as:

$$G = (\mathcal{V}, \mathcal{E}), \tag{2}$$

where the node set $\mathcal{V} = \{v_1, \cdots, v_N\}$ denotes the data examples and $\mathcal{E} \subseteq \{(v_i, v_j) | v_i, v_j \in \mathcal{V}\}$ is a collection of node pairs. Let $\mathcal{N}_i$ be the neighbor set of instances $v_i$ in the proximity graph. To construct $\mathcal{N}_i$, we first form a candidate set $\mathcal{C}_i = \{v_m\}$ for each instance $v_i$ by uniformly picking $M(M \ll N)$ neighbor candidates. Then we select the $K$ nearest ones from the candidate set:

$$\mathcal{N}_i = \operatorname*{TopK}_{v_m \in \mathcal{C}_i} (\mathbf{e}_i \cdot \mathbf{e}_m), \tag{3}$$

where $\cdot$ is the inner product operation. $M$ controls the similarity between the center node and its immediate neighbor nodes, which can be demonstrated by the following proposition:

**Proposition 1.** *Given an observation $v_i$ with the corresponding representation $\mathbf{e}_i$, assume that there are at least $S$ observations whose inner product similarity with $v_i$ is larger than $s$, i.e.,*

$$\left| \left\{ v_j \in \mathcal{V} \mid \mathbf{e}_i \cdot \mathbf{e}_j > s \right\} \right| \geq S. \tag{4}$$

*Then in the proximity graph $G$, the similarity between $v_i$ and its neighbors is larger than $s$ with proximate probability at least:*

$$\mathbb{P} \left\{ \mathbf{e}_i \cdot \mathbf{e}_k > s, \forall v_k \in \mathcal{N}_i \right\} \gtrapprox \left( 1 - p^M \right)^K, \tag{5}$$

*where $p = \frac{N-S}{N}$, and $K$ is the number of neighbors.*

The proof is deferred to Appendix B. The insight of Proposition 1 is to relate candidate set size $M$ to the similarity between a node pair. Higher $M$ indicates a greater probability that two adjacent nodes are similar, and proximity graph will be more like the kNN graph. On the other hand, if $M$ is low, some randomly chosen instances are more likely to be neighbors, improving the diversity of the negatives around the center node.

**Proximity graph sampling.** Breadth-first Sampling (BFS) and Depth-first Sampling (DFS) are two straightforward graph sampling methods (Grover and Leskovec, 2016), representing extreme scenarios in terms of the search space:

- Breadth-first Sampling (BFS) collects all of the current node's immediate neighbors, then moves to its neighbors and repeats the procedure until the number of collected instances reaches batch size.
- Depth-first Sampling (DFS) randomly explores the node branch as far as possible before the number of the visited nodes reaches batch size.

To flexibly explore the negatives in proximity graph, we propose to apply Random Walk with Restart (RWR) which can exhibit a mixture of both. As shown in Algorithm 3, beginning at a node,

---

**Algorithm 1:** In-batch Contrastive Framework with ProSampler

---

**Input:** Dataset $\mathcal{D} = \{x_i | i = 1, \cdots, N\}$, Encoder $f(\cdot)$, Batchsize $B$, Graph update step $t$,
      Modality-specific augmentation functions $\mathcal{T}$.

**for** $iter \leftarrow 0, 1, \cdots$ **do**

> // ProSampler
> **if** $iter \% t == 0$ **then**
>> // Proximity Graph Construction
>> Build the proximity graph $G$ by Algorithm 2.
>
> **end**
> // Proximity Graph Sampling
> Randomly select a start node and get the mini-batch $\{x_i\}_B$ by Algorithm 3.
>
> Obtain positive pairs $\{(x_i, x_i^+)\}_B$ by augmentation functions $f_{aug}(\cdot) \sim \mathcal{T}$.
> Generate representations $\{(\mathbf{e}_i, \mathbf{e}_i^+)\}_B$ by Encoder $f(\cdot)$.
> Compute the loss by Eq. 1, where $\{(\mathbf{e}_i, \mathbf{e}_j)\}_{B(B-1)}^{i \neq j}$ are treated as negative pairs.
> Update the parameters of $f(\cdot)$.

**end**

---

the sampler iteratively teleports back to the start point with probability $\alpha$ or travels to a neighbor of the current position with the probability proportional to the edge weight. The process continues until it collects a fixed number of vertices which will be taken as the sampled batch.

The key insight of using RWR is that it can modulate the probability of sampling within a neighborhood by adjusting $\alpha$, which can be demonstrated by the following proposition:

**Proposition 2.** *For all $0 < \alpha \leq 1$ and $\mathcal{S} \subset \mathcal{V}$, the probability that a Lazy Random Walk with Restart starting from a node $u \in \mathcal{S}$ escapes $\mathcal{S}$ satisfies $\sum_{v \in (\mathcal{V} - \mathcal{S})} \mathbf{p}_u(v) \leq \frac{1-\alpha}{2\alpha} \Phi(\mathcal{S})$, where $\mathbf{p}_u$ is the stationary distribution, and $\Phi(\mathcal{S})$ is the graph conductance of $\mathcal{S}$.*

The proof is deferred to Appendix B. Proposition 2 indicates that the probability of RWR escaping from a local cluster (Andersen et al., 2006; Spielman and Teng, 2013) can be bounded by the graph conductance (Šíma and Schaeffer., 2006) and the restart probability $\alpha$. In other words, higher $\alpha$ indicates that the walker will approximate BFS behavior and sample within a small locality. Lower $\alpha$ encourages the walker to visit the nodes which are further away from the center node.

**ProSampler pipeline.** As shown in Algorithm 1, ProSampler serves as a mini-batch sampler and can be easily plugged into any in-batch contrastive learning method, such as SimCLR (Chen et al., 2020), MoCo v3 (Chen et al., 2021), SimCSE (Gao et al., 2021) and GraphCL (You et al., 2020). Specifically, during the training process, ProSampler first constructs the proximity graph, which will be updated after $t$ training steps, then selects a start node at random and samples a mini-batch on proximity graph by RWR. ProSampler is orthogonal to the contrastive methods.

As shown in Figure 3, the number of candidates $M$ and the restart probability $\alpha$ are the key to flexibly control the hardness of a sampled batch. When we set $M$ as the size of dataset and $\alpha$ as 1, proximity graph is equivalent to kNN graph and graph sampler will only collect the immediate neighbors around a center node, which behaves similarly to a kNN Sampler. On the other hand, if $M$ is set to 1 and $\alpha$ is set to 0, the RWR degenerates into the DFS and chooses the neighbors that are linked at random, which indicates that ProSampler performs as a Uniform Sampler. We provide an empirical criterion of choosing $M$ and $\alpha$ in Section 4.3.

**Complexity.** The time complexity of building a proximity graph is $O(NMd)$ where $N$ is the dataset size, $M$ is the candidate set size and $d$ denotes the embedding size. It is practically efficient since usually $M$ is much smaller than $N$, and the process can be accelerated by embedding retrieval libraries such as Faiss (Johnson et al., 2019). More analysis on efficiency can be found in Appendix F.5. Besides, the space cost of ProSampler mainly comes from graph construction and graph storage. The total space complexity of ProSampler is $O(Nd + NK)$ where $K$ is the number of neighbors in the proximity graph.

## 4 EXPERIMENTS

To show the effectiveness of ProSampler in a variety of scenarios, we apply it to the representative contrastive learning algorithms on three data modalities, including image, text, and graph. Furthermore, to investigate ProSampler's generality, we equip two variants of InfoNCE objective with our model, including DCL (Chuang et al., 2020) and HCL (Robinson et al., 2021). InfoNCE objective and its variants are described in Appendix C. The statistics of the datasets are summarized in Appendix D, and the detailed experimental setting can be found in Appendix E.

### 4.1 BENCHMARKING RESULTS

**Results on Image Modality.** We first adopt Sim-CLR (Chen et al., 2020) and MoCo v3 (Chen et al., 2021) as the backbone based on ResNet-50 (He et al., 2016). We start with training the model for 800 epochs with a batch size of 2048 for SimCLR and 4096 for MoCo v3, respectively. We then use linear probing to evaluate the representations on ImageNet. We also compare the ProSampler with the two state-of-the-art self-supervised learning methods without negative sampling, including SwAV (Caron et al., 2020) and BYOL (Grill et al., 2020). As shown in Table 1, our proposed model can consistently boost the performance of original SimCLR and MoCo v3, and outperforms all the baselines without negatives, demonstrating the superiority of ProSampler. Besides, we evaluate ProSampler on the other benchmark datasets, which can be found in Appendix F.1.

Table 1: Top-1 accuracy under the linear evaluation with the ResNet-50 backbone on ImageNet.

| Method | 100 ep | 800 ep |
|---|---|---|
| SwAV* | 66.5 | 71.8 |
| BYOL | 66.5 | 74.3 |
| SimCLR | 64.0 | 68.7 |
| w/ ProSampler | **64.7** (↑ 0.7) | **69.2** (↑ 0.5) |
| MoCo v3 | 68.9 | 73.8 |
| w/ ProSampler | **69.5** (↑ 0.6) | **74.2** (↑ 0.4) |

* without multi-crop augmentations.

**Results on Text Modality.** We evaluate ProSampler on learning the sentence representations by SimCSE (Gao et al., 2021) framework with pretrained BERT (Devlin et al., 2018) as backbone. The results of Table 2 suggest that ProSampler consistently improves the baseline models with an absolute gain of 1.09%~2.91% on 7 semantic textual similarity (STS) tasks (Agirre et al., 2012; 2013; 2014; 2015; 2016; Tian et al., 2017; Marelli et al., 2014). Specifically, we observe that when applying DCL and HCL, the performance of the self-supervised language model averagely drops by 2.45% and 3.08% respectively. As shown in Zhou et al. (2022) and Appendix F.2, the pretrained language model offers a prior distribution over the sentences, leading to a high cosine similarity of both positive pairs and negative pairs. So DCL and HCL, which leverage the similarity of positive and negative scores to tune the weight of negatives, are inapplicable because the high similarity scores of positives and negatives will result in homogeneous weighting. However, the hard negatives explicitly sampled by our proposed ProSampler can alleviate it, with an absolute improvement of 1.64% on DCL and 2.64% on HCL. The results of RoBERTa (Liu et al., 2019) are reported in Appendix F.3.

Table 2: Overall performance comparison with different negative sampling methods on STS tasks.

| Method | STS12 | STS13 | STS14 | STS15 | STS16 | STS-B | SICK-R | Avg. |
|---|---|---|---|---|---|---|---|---|
| SimCSE-BERT$_{base}$ | 68.62 | 80.89 | 73.74 | 80.88 | 77.66 | **77.79** | **69.64** | 75.60 |
| w/ ProSampler | **72.37** | **82.08** | **75.24** | **83.10** | **78.43** | 77.54 | 68.05 | **76.69** |
| DCL-BERT$_{base}$ | 65.22 | 77.89 | 68.94 | 79.88 | **76.72** | 73.89 | **69.54** | 73.15 |
| w/ ProSampler | **69.55** | **82.66** | **73.37** | **80.40** | 75.37 | **75.43** | 66.76 | **74.79** |
| HCL-BERT$_{base}$ | 62.57 | 79.12 | 69.70 | 78.00 | 75.11 | 73.38 | 69.74 | 72.52 |
| w/ ProSampler | **66.87** | **81.38** | **72.96** | **80.11** | **77.99** | **75.95** | **70.89** | **75.16** |

**Results on Graph Modality.** We test ProSampler with graph-level classification task based on GraphCL (You et al., 2020) framework, which uses GIN (Xu et al., 2018) as the backbone. Table 3 reports the detailed performance comparison on four benchmark datasets: IMDB-B, IMDB-M, COLLAB, and REDDIT-B (Yanardag and Vishwanathan, 2015). The result shows that ProSampler can consistently boost the performance of GraphCL, with an average absolute improvement of

0.89% across all the datasets. Besides, equipped with ProSampler, DCL and HCL can achieve better performance in 6 out of 8 cases. It can also be found that ProSampler can reduce variance in most cases, demonstrating that the hard negatives exploited by ProSampler can enforce the model to learn more robust representations.

Table 3: Accuracy on graph classification task under LIBSVM (Chang and Lin, 2011) classifier.

| Method | IMDB-B | IMDB-M | COLLAB | REDDIT-B |
|---|---|---|---|---|
| GraphCL | 70.90±0.53 | 48.48±0.38 | 70.62±0.23 | 90.54±0.25 |
| w/ ProSampler | **71.90±0.46** | **48.93±0.28** | **71.48±0.28** | **90.88±0.16** |
| DCL | 71.07±0.36 | 48.93±0.32 | **71.06±0.51** | 90.66±0.29 |
| w/ ProSampler | **71.32±0.17** | **48.96±0.25** | 70.44±0.35 | **90.73±0.34** |
| HCL | **71.24±0.36** | 48.54±0.51 | 71.03±0.45 | 90.40±0.42 |
| w/ ProSampler | 71.20±0.38 | **48.76±0.39** | **71.70±0.35** | **91.25±0.25** |

## 4.2 WHY PROSAMPLER PERFORMS BETTER?

In this section, we apply SimCLR on CIFAR10 and CIFAR100, and compare the Uniform Sampler, kNN Sampler and ProSampler in terms of performance and the false negatives to deepen the understanding of ProSampler. We show the performance of ProSampler with different $M$ and $\alpha$ settings in Figure 3, and use the name convention ProSampler $(M, \alpha)$. Besides, we illustrate the histogram of cosine similarity for all pairs from a sampled batch, and the percentage of false negatives within the mini-batch during training in Figure 4. We can observe that although kNN Sampler can explicitly draw a data batch with similar pairs, it introduces a substantially higher number of false negatives, degrading performance significantly. Uniform Sampler is independent of the model so the percentage of FN within the sampled batch remains consistent during training. However, ProSampler can modulate $M, \alpha$ to find the best balance between these two sampling methods. We can observe that ProSampler can sample hard mini-batch but only exhibits a slightly higher percentage of false negatives than Uniform Sampler with optimal parameter setting, which enables ProSampler to achieve the best performance. Similar phenomenon on CIFAR100 can be found in Appendix F.4.

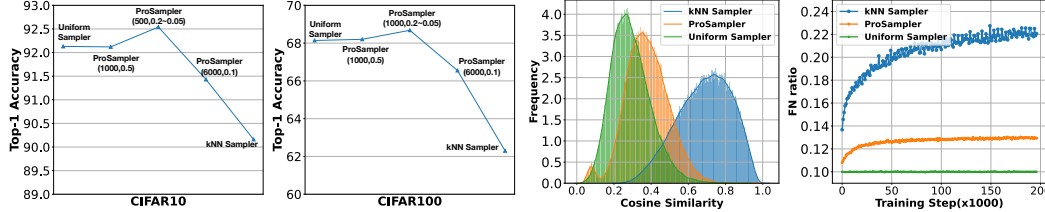

Figure 3: Performance comparison of different in-batch samplers on image classification task.

Figure 4: Cosine similarity and false negative ratio on CIFAR10 dataset.

## 4.3 EMPIRICAL CRITERION FOR PROSAMPLER

To analyze the impact of the size of neighbor candidate $M$ and the random walk restart probability $\alpha$, we vary the $M$ and $\alpha$ in range of $\{500, 1000, 2000, 4000, 6000\}$ and $\{0.1, 0.3, 0.5, 0.7\}$ respectively, and apply SimCLR, SimCSE and GraphCL as backone. We summarize the results in Table 4 and Table 5. Table 4 shows that in most cases, the performance of the model peaks when $M = 1000$ but plumbs quickly with the increasing of $M$. Such phenomena are consistent with the intuition that higher $M$ raises the probability of selecting similar instances as neighbors, but the sampler will be more likely to draw the mini-batch with false negatives, degrading the performance.

Table 5 shows the performance of ProSampler with different $\alpha$. Besides, to better understand the effect of $\alpha$, we illustrate the histograms of cosine similarity for all pairs from a sampled batch after training in Figure 5, and plot the percentage of false negatives in the mini-batch during training in Figure 6. We can observe that $\alpha$ moving from 0.1 through 0.2 to 0.7 causes cosine similarities to gradually skew left, but introduces more false negatives in the batch, creating a trade-off. This phenomenon indicates

Table 4: Impact of neighbor candidates $M$.

| $M$ | 500 | 1000 | 2000 | 4000 | 6000 |
|---|---|---|---|---|---|
| CIFAR10 | **92.54** | 92.49 | 91.83 | 91.72 | 91.43 |
| CIFAR100 | 67.92 | **68.68** | 67.05 | 66.19 | 65.55 |
| STL10 | 84.16 | **84.38** | 82.80 | 81.91 | 80.92 |
| ImageNet-100 | 59.6 | **60.8** | 60.1 | 59.1 | 58.4 |
| Wikipedia | 71.36 | **76.69** | 76.09 | 75.76 | 75.11 |
| COLLAB | 70.47 | **71.48** | 70.93 | 70.46 | 70.24 |

Table 5: Impact of restart probability $\alpha$.

| $\alpha$ | 0.1 | 0.3 | 0.5 | 0.7 | 0.2∼0.05 |
|---|---|---|---|---|---|
| CIFAR10 | 92.41 | 92.26 | 92.12 | 92.06 | **92.54** |
| CIFAR100 | 68.31 | 67.98 | 68.20 | 68.00 | **68.68** |
| STL10 | 83.01 | 80.69 | 83.93 | 82.56 | **84.38** |
| ImageNet-100 | 60.8 | 59.6 | 58.1 | 57.7 | **60.8** |
| Wikipedia | 71.74 | 72.13 | 72.41 | **76.69** | – |
| COLLAB | 70.36 | 70.63 | 70.63 | 70.31 | **71.48** |

that the sampler with a higher $\alpha$ sample more frequently within a local neighborhood, which is more likely to yield similar pairs. However, as training progresses, the instances of the same class tend to group together, increasing the probability of collecting false negatives. To find the best balance, we linearly decay $\alpha$ from 0.2 to 0.05 as the training epoch increases, which is presented as $0.2 \sim 0.05$ in Table 5. It can be found that this dynamic strategy achieves the best performance in all cases except SimCSE which only trains for one epoch. Interestingly, SimCSE achieves the best performance by a large margin when $\alpha = 0.7$ since hard negatives can alleviate the distribution issue brought by the pre-trained language model. More analysis can be found in Section 4.1 and Appendix F.2.

To sum up, the suggested $M$ would be 500 for the small-scale dataset, and 1000 for the larger dataset. The suggested $\alpha$ should be relatively high, e.g., 0.7, for the pre-trained language model-based method. Besides, dynamic decay $\alpha$, e.g., 0.2 to 0.05, is the best strategy for the other methods.

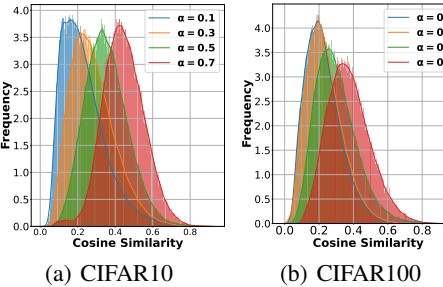

(a) CIFAR10     (b) CIFAR100

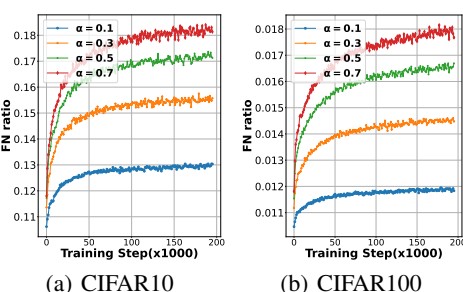

(a) CIFAR10     (b) CIFAR100

Figure 5: Cosine similarities between the pairs.     Figure 6: Percentage of false negatives.

## 4.4 DISCUSSIONS

Due to the page limit, some additional experiments are reported in Appendix F. Appendix F.5 studies the efficiency of ProSampler. Appendix F.6 compares different graph sampling methods in terms of performance, cosine similarity, and false negatives. Appendix F.7 compares the performance of ProSampler with proximity graph and kNN graph. Appendix F.8 discusses the influence of some parameters, including batchsize $B$, neighbor number $K$, and proximity graph update interval $t$. Appendix F.9 presents the training curves. Appendix F.10 includes case studies where we show some real cases of the mini-batch sampled by ProSampler and Uniform Sampler.

## 5 CONCLUSION

In this paper, we study the problem of global hard negative sampling for in-batch contrastive learning. We reformulate the original mini-batch sampling problem to the proximity graph sampling problem. Based on this, we propose a proximity graph-based sampling framework, ProSampler, which can sample a mini-batch with hard negative pairs for in-batch contrastive learning at each training step. Besides, we conduct experiments on three state-of-the-art contrastive methods with different modalities and two variants of InfoNCE objective to evaluate our proposed ProSampler, which shows that ProSampler can consistently improve the these models.

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

# A   ALGORITHM DETAIL

Here we present the detail of Proximity Graph Construction and Random Walk with Restart, as shown in Algorithm 2 and Algorithm 3 respectively.

---

**Algorithm 2:** Proximity Graph Construction

---

**Input:** Dataset $\mathcal{D} = \{x_i\}$, Candidate set size $M$, Neighbor number $K$;
**Output:** A proximity graph $G$;
**for** $v$ *in* $\mathcal{D}$ **do**
  | Randomly select $M$ neighbor candidates from $\mathcal{D}$;
  | Select the $K$ closest candidates $\mathcal{N}_v$ by Eq. 3;
  | $G[v] \leftarrow \mathcal{N}_v$;
**end**
**return** $G$

---

---

**Algorithm 3:** Random Walk with Restart(RWR)

---

**Input:** Proximity graph $G = \{\mathcal{V}, \mathcal{E}\}$, seed node $u$, restart probability $\alpha$, number of sampled node $B$;
**Output:** A sampled node set $\mathcal{S}$;
$\mathcal{S} \leftarrow \{\}, v \leftarrow u$;
**while** *len*$(\mathcal{S}) < B$ **do**
  | **if** $v$ *not in* $\mathcal{S}$ **then**
  |   | $\mathcal{S}$.insert($v$)
  | **end**
  | Sample $r$ from Uniform distribution $U(0, 1)$;
  | **if** $r < \alpha$ **then**
  |   | $v \leftarrow u$;
  | **end**
  | **else**
  |   | Randomly sample $\hat{v}$ from $v$'s neighbors;
  |   | $v \leftarrow \hat{v}$;
  | **end**
**end**
**return** $\mathcal{S}$

---

# B   THEORETICAL PROOF

**Proposition 1.** *Given an observation $v_i$ with the corresponding representation $\mathbf{e}_i$, assume that there are at least $S$ observations whose inner product similarity with $v_i$ is larger than $s$, i.e.,*

$$\left| \left\{ v_j \in \mathcal{V} \mid \mathbf{e}_i \cdot \mathbf{e}_j > s \right\} \right| \geq S. \tag{1}$$

*Then in the proximity graph $G$, the similarity between $v_i$ and its neighbors is larger than $s$ with proximate probability at least:*

$$\mathbb{P}\left\{ \mathbf{e}_i \cdot \mathbf{e}_k > s, \forall v_k \in \mathcal{N}_i \right\} \gtrapprox \left( 1 - p^M \right)^K, \tag{2}$$

*where $p = \frac{N-S}{N}$, and $K$ is the number of neighbors.*

*Proof.* Since $M \ll N$, we can approximately assume that the sampling is with replacement. In this case, we have

$$\mathbb{P}\left\{ \mathbf{e}_i \cdot \mathbf{e}_k > s, \forall v_k \in \mathcal{N}_i \right\} = 1 - \sum_{k=0}^{K-1} \binom{M}{k} p^{M-k} \left( 1 - p \right)^k. \tag{3}$$

Then let us prove (2) by induction. When $K = 1$, the conclusion clearly holds.

Assuming that the conclusion holds when $K = L - 1$, let us consider the case when $K = L$. We have

$$1 - \sum_{k=0}^{L-1} \binom{M}{k} p^{M-k}(1-p)^k \gtrapprox \left(1 - p^M\right)^{L-1} - \binom{M}{L-1} p^{M-L+1}(1-p)^{L-1}. \qquad (4)$$

To prove the conclusion, we only need to show

$$\left(1 - p^M\right)^{L-1} p^M \gtrapprox \binom{M}{L-1} p^{M-L+1}(1-p)^{L-1}, \qquad (5)$$

or equivalently

$$\left(1 - p^M\right)^{L-1} p^{L-1} = \left(1 - p^M\right)^{L-1} \left(\frac{N-S}{N}\right)^{L-1}$$
$$\gtrapprox \binom{M}{L-1} \left(\frac{S}{N}\right)^{L-1} = \binom{M}{L-1}(1-p)^{L-1}. \qquad (6)$$

On the other hand, according to Knuth (1997), we have

$$\binom{M}{L-1} \leq \left(\frac{eM}{L-1}\right)^{L-1}, \qquad (7)$$

where $e$ denotes the Euler's number. Substituting (7) into (6), we only need to show

$$(N-S)(L-1)\left(1 - p^M\right) \gtrapprox eMS. \qquad (8)$$

The above relation holds depending on the choices of $M$, $S$ and $L$, which can be approximately satisfied in our scenario. $\qquad \square$

**Proposition 2.** *For all $0 < \alpha \leq 1$ and $\mathcal{S} \subset \mathcal{V}$, the probability that a Lazy Random Walk with Restart starting from a node $u \in \mathcal{S}$ escapes $\mathcal{S}$ satisfies $\sum_{v \in (\mathcal{V} - \mathcal{S})} \mathbf{p}_u(v) \leq \frac{1-\alpha}{2\alpha} \Phi(\mathcal{S})$, where $\mathbf{p}_u$ is the stationary distribution, and $\Phi(\mathcal{S})$ is the graph conductance of $\mathcal{S}$.*

*Proof.* We first introduce the definition of graph conductance (Šíma and Schaeffer., 2006) and Lazy Random Walk (Spielman and Teng, 2013):

**Graph Conductance** . For an undirected graph $G = (\mathcal{V}, \mathcal{E})$, the graph volume of a node set $\mathcal{S} \subset \mathcal{V}$ is defined as $\text{vol}(\mathcal{S}) = \sum_{v \in \mathcal{S}} d(v)$, where $d(v)$ is the degree of node $v$. The edge boundary of a node set is defined to be $\partial(\mathcal{S}) = \{(x, y) \in \mathcal{E} | x \in \mathcal{S}, y \notin \mathcal{S}\}$. The conductance of $\mathcal{S}$ is calculated as followed:

$$\Phi(\mathcal{S}) = \frac{|\partial(\mathcal{S})|}{\min(\text{vol}(\mathcal{S}), \text{vol}(\mathcal{V} - \mathcal{S}))} \qquad (9)$$

**Lazy Random Walk** . Lazy Random Walk (LRW) is a variant of Random Walk, which first starts at a node, then stays at the current position with a probability 1/2 or travels to a neighbor. The transition matrix of a lazy random walk is $\mathbf{M} \triangleq (\mathbf{I} + \mathbf{A}\mathbf{D}^{-1})/2$, where the $\mathbf{I}$ denotes the identity matrix, $\mathbf{A}$ is the adjacent matrix, and $\mathbf{D}$ is the degree matrix. The $K$-th step Lazy Random Walk distribution starting from a node $u$ is defined as $\mathbf{q}^{(K)} \leftarrow \mathbf{M}^K \mathbf{1}_u$.

We then present a theorem which relates the Lazy Random Walk to graph conductance, which has been proved in Spielman and Teng (2013):

**Theorem 1.** *For all $K \geq 0$ and $\mathcal{S} \subset \mathcal{V}$, the probability that a $K$-step Lazy Random Walk starting at $u \in \mathcal{S}$ escapes $\mathcal{S}$ satisfies $\mathbf{q}^{(K)}(\mathcal{V} - \mathcal{S}) \leq K\Phi(\mathcal{S})/2$.*

Theorem 1 guarantees that given a non-empty node set $\mathcal{S} \subset \mathcal{V}$ and a start node $u \in \mathcal{S}$, the Lazy Random Walker will be more likely stuck at $\mathcal{S}$. Here we extend the LRW to Lazy Random Walk with Restart (LRWR) which will return to the start node with probability $\alpha$ or perform Lazy Random

Walk. According to the previous studies (Page et al., 1999; Avrachenkov et al., 2014; Chung and Zhao, 2010; Tong et al., 2006), we can obtain a stationary distribution $\mathbf{p}_u$ by recursively performing Lazy Random Walk with Restart, which can be formulated as a linear system:

$$\mathbf{p}_u = \alpha \mathbf{1}_u + (1 - \alpha)\mathbf{M}\mathbf{p}_u \tag{10}$$

where $\alpha$ denotes the restart probability. The element $\mathbf{p}_u(v)$ represents the probability of the walker starting at $u$ and ending at $v$. $\mathbf{p}_u$ can be expressed by a geometric sum of Lazy Random Walk (Chung and Tsiatas, 2010):

$$\mathbf{p}_u = \alpha \sum_{l=0}^{\infty} (1 - \alpha)^l \mathbf{M}^l \mathbf{1}_u = \alpha \sum_{l=0}^{\infty} (1 - \alpha)^l \mathbf{q}_u^{(l)} \tag{11}$$

Applying the Theorem 1, we have:

$$\sum_{v \in (\mathcal{V}-\mathcal{S})} \mathbf{p}_u(v) = \alpha \sum_{l=0}^{\infty} \sum_{v \in (\mathcal{V}-\mathcal{S})} (1-\alpha)^l \mathbf{q}_u^{(l)}(v) \leq \alpha \sum_{l=0}^{\infty} l(1-\alpha)^l \Phi(\mathcal{S})/2 = \frac{1-\alpha}{2\alpha}\Phi(\mathcal{S}) \tag{12}$$

The desired result is obtained by comparing the two sides of (12). □

In particular, the only difference between Lazy Random with Restart and Random Walk with Restart is that the former has a probability of remaining in the current position without taking any action. They are equivalent when sampling a predetermined number of nodes.

## C INFoNCE OBJECTIVE AND ITS VARIANTS

Here we describe in detail the objective functions of three in-batch contrastive learning methods, including SimCLR (Chen et al., 2020), GraphCL (You et al., 2020) and SimCSE (Gao et al., 2021). Besides, we cover two variants, i.e., DCL (Chuang et al., 2020) and HCL (Robinson et al., 2021), which are also applied in the experiments.

### C.1 SIMCLR

SimCLR (Chen et al., 2020) first uniformly draws a mini-batch of instances $\{x_1 ... x_B\} \subset \mathcal{D}$, then augments the instances by two randomly sampled augmentation strategies $f_{aug}(\cdot), f'_{aug}(\cdot) \sim \mathcal{T}$, resulting in $2B$ data points. Two augmented views $(x_i, x_{i+B})$ of the same image are treated as a positive pair, while the other $2(B-1)$ examples are negatives. The objective function applied in SimCLR for a positive pair $(x_i, x_{i+B})$ is formulated as:

$$\ell_{i,i+B} = -\log \frac{e^{f(x_i)^T f(x_{i+B})/\tau}}{\sum_{j \neq i}^{2B} e^{f(x_i)^T f(x_j)/\tau}}, \tag{13}$$

where $\tau$ is the temperature and $f(\cdot)$ is the encoder. The loss is calculated for all positive pairs in a mini-batch, including $(x_i, x_{i+B})$ and $(x_{i+B}, x_i)$. It can be found that SimCLR takes all $2(B-1)$ augmented instances within a mini-batch as negatives.

### C.2 GRAPHCL AND SIMCSE

Similar as SimCLR, the objective function of GraphCL (You et al., 2020) and SimCSE (Gao et al., 2021) is defined on the augmented instance pairs within a mini-batch. Given a sampled mini-batch $\{x_1 ... x_B\} \subset \mathcal{D}$, both GraphCL and SimCSE apply data augmentation to obtain positive pairs, and the loss function for a positive pair $(x_i, x_i^+)$ can be formulated as:

$$\ell_i = -\log \frac{e^{f(x_i)^T f(x_i^+)/\tau}}{\sum_{j=1}^{B} e^{f(x_i)^T f(x_j^+)/\tau}}. \tag{14}$$

Compared with the SimCLR, GraphCL and SimCSE only take the other $B-1$ augmented instances as negatives.

### C.3 DCL AND HCL

DCL (Robinson et al., 2021) and HCL (Robinson et al., 2021) are two variants of InfoNCE objective function, which aim to alleviate the false negative issue or mine the hard negatives by reweighting the negatives in the objective. The main idea behind them is using the positive distribution to correct for the negative distribution.

For simplicity, we annotate the positive score $e^{f(x_i)^T f(x_i^+)/\tau}$ as $pos$, and negative score $e^{f(x_i)^T f(x_j^+)/\tau}$ as $neg_{ij}$. Given a mini-batch and a positive pair $(x_i, x_i^+)$, the reweighting negative distribution proposed in DCL and HCL are:

$$\max \left( \sum_{j=1}^{B} \frac{-N_{neg} \times \tau^+ \times pos + \lambda_{ij} \times neg_{ij}}{1 - \tau^+}, e^{-1/\tau} \right), \tag{15}$$

where $N_{neg}$ is the number of the negatives in mini-batch, $\tau^+$ is the class probability, $\tau$ is the temperature, and $\lambda_{ij}$ is concentration parameter which is simply set as 1 in DCL or calculated as $\lambda_{ij} = \frac{\beta \times neg_{ij}}{\sum neg_{ij}/N_{neg}}$ in HCL. All of $\tau^+, \tau, \beta$ are tunable hyperparameters. The insight of Eq.15 is that the negative pair with the score closer to positive score will be assigned lower weight in loss function. In other words, the similarity difference between positive and negative pairs dominates the weighting function.

## D DATASET DETAILS

For image representation learning, we adopt five benchmark datasets, comprising of CIFAR10, CIFAR100, STL10, ImageNet-100 and ImageNet ILSVRC-2012 (Russakovsky et al., 2015). Information on the statistics of these datasets is summarized in Table 6. For graph-level representation learning, we conduct experiments on IMDB-B, IMDB-M, COLLAB and REDDIT-B (Yanardag and Vishwanathan, 2015), the details of which are presented in Table 7. For text representation learning, we evaluate the method on a one-million English Wikipedia dataset which is used in the SimCSE and can be downloaded from HuggingFace repository[1].

Table 6: Statistics of datasets for image classification task.

| Datasets | CIFAR10 | CIFAR100 | STL10 | ImageNet-100 | ImageNet |
|---|---|---|---|---|---|
| #Train | 50,000 | 50,000 | 105,000 | 130,000 | 1,281,167 |
| #Test | 10,000 | 10,000 | 8,000 | 50,00 | 50,000 |
| #Classes | 10 | 100 | 10 | 100 | 1,000 |

## E EXPERIMENTAL DETAILS

### E.1 IMAGE REPRESENTATIONS

In image domain, we apply SimCLR (Chen et al., 2020) and MoCo v3 (Chen et al., 2021) as the baseline method, with ResNet-50 (He et al., 2016) as an encoder to learn image representations. The

---

[1] https://huggingface.co/datasets/princeton-nlp/datasets-for-simcse/resolve/main/wiki1m_for_simcse.txt

Table 7: Statistics of datasets for graph-level classification task.

| Datasets | IMDB-B | IMDB-M | COLLAB | REDDIT-B |
|---|---|---|---|---|
| #Graphs | 1,000 | 1,500 | 5,000 | 2,000 |
| #Classes | 2 | 3 | 3 | 2 |
| Avg. #nodes | 19.8 | 13.0 | 74.5 | 429.7 |

feature map generated by ResNet-50 block is projected to a 128-D image embedding via a two-layer MLP (2048-D hidden layer with ReLU activation function). Besides, the output vector is normalized by $l_2$ normalization (Wu et al., 2018). We employ two sampled data augmentation strategies to generate positive pairs, and implicitly use other examples in the same mini-batch as negative samples.

For CIFAR10, CIFAR100 and STL10, all models are trained for 1000 epochs with the default batch size $B$ of 256. We use the Adam optimizer (Kingma and Ba, 2015) with learning rate of 0.001 for optimization. The temperature parameter is set as 0.5 and the dimension of image embedding is set as 128. For ImageNet-100 and ImageNet, we train the models with 100 and 400 epochs respectively, and use LARS optimizer (You et al., 2019) with learning rate of $0.3 \times B/256$ and weight decay of $10^{-6}$. Here, the batch size is set as 2048 for ImageNet and 512 for ImageNet-100, respectively. We fix the temperature parameter as 0.1 and the image embedding dimension as 128. After the unsupervised learning, we train a supervised linear classifier for 100 epochs on the top of the frozen learned representations.

As for ProSampler, we update the proximity graph per 100 training iterations. We fix the number of neighbors $K$ as 100 for CIFAR10, CIFAR100 and STL10. The size of neighbor candidate set $M$ is set as 1000 for CIFAR100 and STL10, and 500 for CIFAR10. Besides, the initial restart probability $\alpha$ of RWR (Random Walk with Restart) is set to 0.2 and decays linearly to 0.05 with the training process. For ImageNet-100 and ImageNet, we keep $M$ as 1000 and $K$ as 500. The restart probability $\alpha$ is fixed as 0.1.

### E.2 GRAPH REPRESENTATIONS

In graph domain, we use the GraphCL (You et al., 2020) framework as the baseline and GIN (Xu et al., 2018) as the backone. We run ProSampler 5 times with different random seeds and report the mean 10-fold cross-validation accuracy with variance. We apply Adam optimizer (Kingma and Ba, 2015) with a learning rate of 0.01, and 3-layer GIN with a fixed hidden size of 32. We set the temperature as 0.2 and gradually decay the restart probability of RWR ($0.2 \sim 0.05$). Proximity graph will be updated after $t$ iterations. The overall hyperparameter settings on different datasets are summarized in Table 8.

Table 8: Hyperparameter settings for graph-level representation learning.

| Datasets | IMDB-B | IMDB-M | COLLAB | REDDIT-B |
|---|---|---|---|---|
| Batchsize | 256 | 128 | 128 | 128 |
| Epoch | 100 | 50 | 20 | 50 |
| $t$ | 25 | 25 | 50 | 50 |
| $M$ | 300 | 300 | 1,000 | 500 |
| $K$ | 100 | 100 | 100 | 100 |

### E.3 TEXT REPRESENTATIONS

In text domain, we use SimCSE (Gao et al., 2021) as baseline method and adopt the pretrained BERT and RoBERTa provided by HuggingFace[2] for sentence embedding learning. Following the training setting of SimCSE, we train the model for one epoch in an unsupervised manner and evaluate it

---

[2]https://huggingface.co/models

on 7 STS tasks. Proximity graph will be only built once based on the pretrained language models before training. For BERT, we set the batch size to 64 and learning rate to $3 \times 10^{-5}$. For RoBERTa, the batch size is set as 512 and learning rate is fixed as $10^{-5}$. We keep the temperature as 0.05, the number of neighbor candidates $M$ as 1000, the number of neighbors $K$ as 500, and the restart probability $\alpha$ as 0.7 for both BERT and RoBERTa.

# F   ADDITIONAL EXPERIMENTS

## F.1   EXTENSIVE STUDIES ON COMPUTER VISION

Here we evaluate the ProSampler on two small-scale (CIFAR10,CIFAR100) and two medium-scale (STL10,ImageNet-100) benchmark datasets, and equip DCL (Chuang et al., 2020) and HCL (Robinson et al., 2021)[3] with ProSampler to investigate its generality. Experimental results in Table 9 show that ProSampler can consistently improve SimCLR and its variants on all the datasets, with an absolute gain of 0.3%~2.5%. We also can observe that the improvement is greater on medium-scale datasets than on small-scale datasets. Specifically, the model equipped with HCL and ProSampler achieves a significant improvement (6.23%) on STL10 over the original SimCLR.

Table 9: Overall performance comparison on image classification task in term of Top-1 Accuracy.

| Method | CIFAR10 | CIFAR100 | STL10 | ImageNet-100 |
|---|---|---|---|---|
| SimCLR | 92.13 | 68.14 | 83.26 | 59.30 |
| w/ ProSampler | **92.54** | **68.68** | **84.38** | **60.80** |
| DCL | 92.28 | 68.52 | 84.92 | 59.90 |
| w/ ProSampler | **92.74** | **68.91** | **86.39** | **60.14** |
| HCL | 92.39 | 68.92 | 88.20 | 60.60 |
| w/ ProSampler | **92.41** | **69.13** | **89.49** | **61.50** |

## F.2   SIMILARITY COMPARISON BETWEEN POSITIVE AND NEGATIVE PAIRS

To explain the performance degradation of DCL and HCL objectives, we select 12 representative mini-batches and plot the cosine similarity histogram of positive and negative pairs on BERT (top) and RoBERTa (bottom) in Figure 7. We observe the following: (1) At the start of and throughout the training, the positive pairs are assigned a high cosine similarity (around 0.9) by the pretrained language model; (2) The negative similarities begin with a relative high score and gradually skew left because of the self-supervised learning. Such phenomenon is consistent to Zhou et al. (2022). DCL and HCL which leverage the difference between positive and negative similarity to reweight the negative scores are inapplicable since the low distribution gap between positive and negative similarities will lead to homogeneous weighting in the objective.

## F.3   TEXT REPRESENTATIONS WITH ROBERTA

We also apply ProSampler to the SimCSE with the pretrained RoBERTa, and present the results in Table 10. Similar as the results of BERT, ProSampler can consistently improve the performance of the baseline model. Besides, as discussed in Section 4.1 and Section F.2, the hard negative sampled by ProSampler explicitly can alleviate the low distribution gap between positive score and negative score distribution caused by the pretrained language model, alleviating the performance degradation of DCL and HCL.

---

[3]DCL and HCL are more like variants of InfoNCE loss, which adjust the weights of negative samples in the original InfoNCE loss.

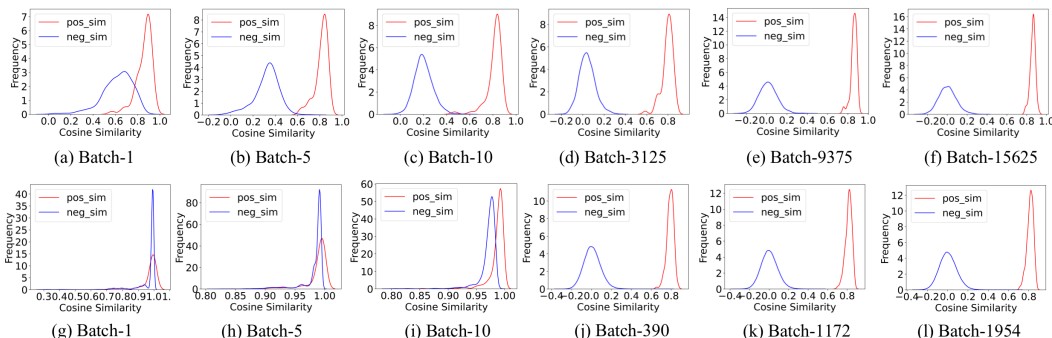

Figure 7: Histograms of cosine similarity on BERT (top) and RoBERTa (bottom).

Table 10: Performance comparison for sentence embedding learning based on RoBERTa.

| Method | STS12 | STS13 | STS14 | STS15 | STS16 | STS-B | SICK-R | Avg. |
|---|---|---|---|---|---|---|---|---|
| SimCSE-RoBERTa$_{base}$ | 67.90 | 80.91 | 73.14 | 80.58 | 80.74 | 80.26 | 69.87 | 76.20 |
| w/ ProSampler | **68.29** | **81.96** | **73.86** | **82.16** | **80.94** | **80.77** | 69.30 | **76.75** |
| DCL-RoBERTa$_{base}$ | **66.60** | 79.16 | **71.05** | 80.40 | 77.76 | **77.94** | **67.57** | 74.35 |
| w/ ProSampler | 65.53 | **80.09** | 71.00 | **80.64** | **78.35** | 77.75 | 67.52 | **74.41** |
| HCL-RoBETa$_{base}$ | **67.20** | 80.47 | 72.44 | 80.88 | 80.57 | 78.79 | 67.98 | 75.49 |
| w/ ProSampler | 66.01 | **80.79** | **73.58** | **81.25** | **80.66** | **79.22** | **68.52** | **75.72** |

## F.4 Cosine Similarity and False Negative Ratio on CIFAR100

Here we compare the Uniform Sampler, kNN Sampler, and ProSampler in terms of cosine similarity and false negatives on CIFAR100. Specifically, we show the histogram of cosine similarity for all pairs in a sampled batch, and the false negative ratio of the mini-batch in Figure 8. It can be found that ProSampler exhibits a balance of Uniform Sampler and kNN Sampler, which can sample the hard negative pair but only brings slightly greater number of false negatives than Uniform Sampler. More analysis can be found in Section 4.2.

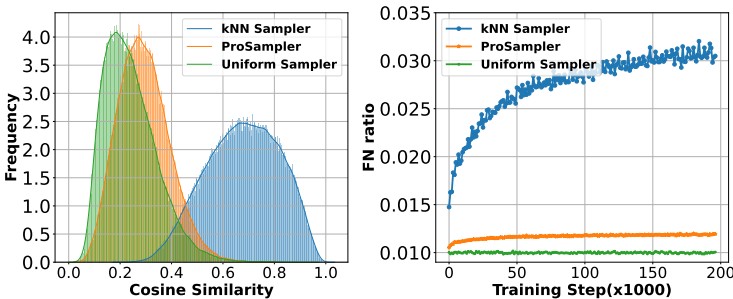

Figure 8: Cosine similarity and false negative ratio on CIFAR100.

## F.5 Efficiency Analysis

To further investigate the efficiency of ProSampler, we analyze the wall-clock time performance. Here, we introduce three metrics to analyze the time cost of mini-batch sampling by ProSampler: (1) **Batch Sampling Cost** ($\text{Cost}_S$) is the average time of RWR taken to sample a mini-batch from a proximity graph; (2) **Proximity Graph Construction Cost** ($\text{Cost}_G$) refers to the time consumption of ProSampler for constructing a proximity graph; (3) **Batch Training Cost** ($\text{Cost}_T$) is the average time taken by the encoder to forward and backward; (4) **Proximity Graph Construction Amortized**

**Cost** ($\text{Cost}_{G/t}$) is the ratio of $\text{Cost}_G$ to the graph update interval $t$. The time cost of ProSampler is shown in Table 11, from which we make the following observations: (1) Sampling a mini-batch $\text{Cost}_S$ takes an order of magnitude less time than training with a batch $\text{Cost}_T$ at most cases. (2) Although it takes 100s for ProSampler to construct a proximity graph in ImageNet, the cost shares across $t$ training steps, which take only $\text{Cost}_{G/t} = 0.2$ per batch. A similar phenomenon can be found in the other datasets as well. In particular, SimCSE only trains for one epoch, and proximity graph is only built once.

Table 11: Time cost of mini-batch sampling by ProSampler on a NVIDIA V100 GPU.

| Metric | STL10 | ImageNet-100 | Wikipedia | ImageNet |
|---|---|---|---|---|
| $\text{Cost}_S$ | 0.013s | 0.015s | 0.005 | 0.15s |
| $\text{Cost}_G$ | 2s | 3s | 79s | 100s |
| $\text{Cost}_T$ | 0.55s | 1.1s | 0.08s | 1.1s |
| $\text{Cost}_{G/t}$ | $0.02(t=100)$ | $0.03(t=100)$ | $0.005(t=15625)$ | $0.2(t=500)$ |

## F.6 COMPREHENSIVE ANALYSIS ABOUT STRATEGIES OF PROXIMITY GRAPH SAMPLING

We conduct an experiment to explore different choices of graph sampling methods, including (1) **Depth First Search** (DFS); (2) **Breadth First Search** (BFS); (3) **Random Walk** (RW); (4) **Random Walk with Restart** (RWR). Table 12 presents an overall performance comparison with different graph sampling methods. Besides, we illustrate the histograms of cosine similarity for all pairs from a sampled batch after finishing training and plot the percentage of false negatives in the mini-batch during training in Figure 9. It can be observed that although BFS brings the most similar pairs in the mini-batch, it performs worse than the original SimCLR since it introduces substantial false negatives. While having a slightly lower percentage of false negatives than RWR, DFS and RW do not exhibit higher performance since they are unable to collect the hard negatives in the mini-batch. The restart property allows RWR to exhibit a mixture of DFS and BFS, which can flexibly modulate the hardness of the sampled batch and find the best balance between hard negatives and false negatives. Benefiting from it, RWR achieves the best performance over the other sampling methods.

Table 12: Overall performance comparison with different graph sampling methods.

| Method | BFS | DFS | RW | RWR |
|---|---|---|---|---|
| CIFAR10 | 91.03 | 92.14 | 92.28 | **92.54** |
| CIFAR100 | 65.15 | 68.29 | 68.33 | **68.68** |
| STL10 | 77.08 | 83.05 | 83.54 | **84.38** |

Table 13: The performance comparison of different batchsize $B$.

| $B$ | 16 | 32 | 64 | 128 | 256 |
|---|---|---|---|---|---|
| CIFAR10 | 79.36 | 84.64 | 89.09 | 91.03 | **92.54** |
| CIFAR100 | 46.59 | 56.24 | 61.30 | 65.96 | **68.68** |
| STL10 | 56.31 | 68.61 | 74.24 | 82.56 | **84.38** |

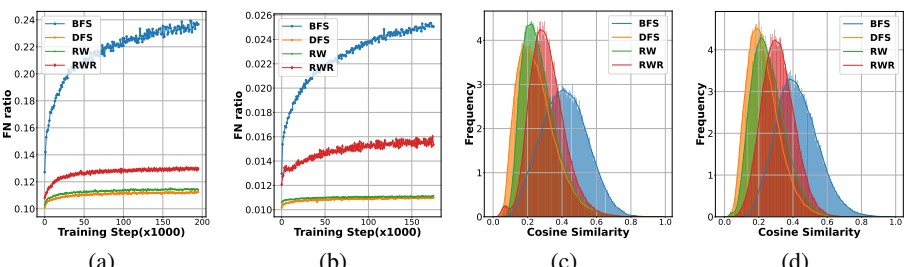

(a)     (b)     (c)     (d)

Figure 9: **(a,b)** Percentage of false negative in a batch sampled from CIFAR10 and CIFAR100 over the training step using different graph sampling methods. **(c,d)** Histograms of cosine similarity of all pairs in a batch for embeddings trained on CIFAR10 and CIFAR100 using different sampling methods.

## F.7 COMPARISON BETWEEN PROXIMITY GRAPH AND KNN GRAPH

To demonstrate the effectiveness of proximity graph, we do an ablation study by replacing proximity graph with kNN graph which directly selects $k$ neighbors with the highest scores for each instance from the whole dataset. The neighbor number $k$ is 100 by default. The comparison results are shown in Table 14, from which we can observe that proximity graph outperforms the kNN graph by a margin. ProSampler with kNN graph even performs worse than the original contrastive learning method because of the false negatives.

| Method | SimCLR | kNN graph | proximity graph |
|---|---|---|---|
| CIFAR-10 | 92.13 | 90.47 | 92.54 |
| CIFAR-100 | 68.14 | 62.67 | 68.68 |

Table 14: Performance comparison of different graph construction methods.

To develop a intuitive understanding of how Proximity graph alleviates the false negative issue, Figure 10 plots the changing curve of false negative ratio in a batch. The results show that Proximity graph could discard the false negative significantly: by the end of the training, kNN will introduce more than 22% false negatives in a batch, while Proximity graph brings about 13% on the CIFAR10 dataset. Similar phenomenon can also be found on CIFAR100 dataset.

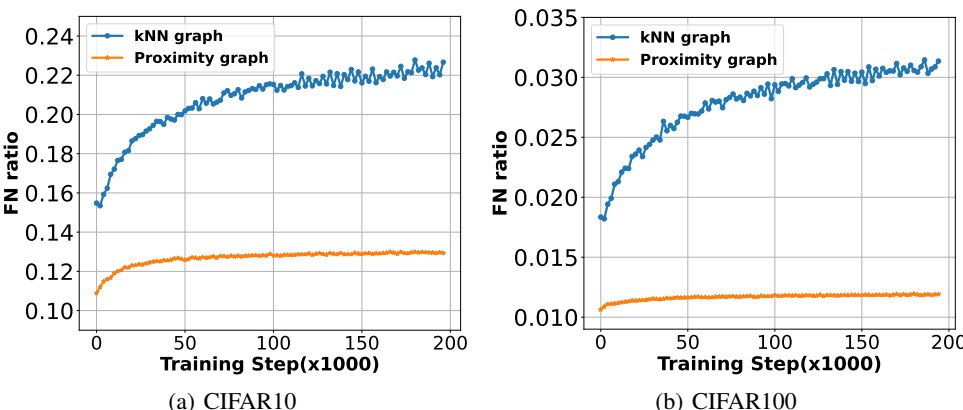

(a) CIFAR10        (b) CIFAR100

Figure 10: Percentage of false negative using different graph building methods over the training step.

## F.8 PARAMETER ANALYSIS

### F.8.1 BATCHSIZE $B$

To analyze the impact of the batchsize $B$, we vary $B$ in the range of {16, 32, 64, 128, 256} and summarize the results in Table 13. It can be found that a larger batchsize leads to better results, which is consistent with the previous studies (Chen et al., 2020; He et al., 2020; Kalantidis et al., 2020).

### F.8.2 IMPACT OF NEIGHBOR NUMBER $K$

In Figure 11, we investigate the impact of the neighbor number $K$ on ImageNet-100 dataset with the default ProSampler setting. We observe that an absolute improvement of 1.1% with the increasing size of neighbors. Specifically, model achieves an absolute performance gain of 0.9% from $K = 100$ to $K = 300$, while only obtains 0.2% from $K = 300$ to $K = 500$. Such experimental results are consistent with our prior philosophy, in which sampling more neighbors always increase the scale of proximity graph and urging ProSampler to explore smaller-scope local cluster (i.e. sample harder negatives within a batch), leading to a significant improvement in performance at first. However, performance degrades after reaching the optimum, because larger $K$ introduces more easy negatives.

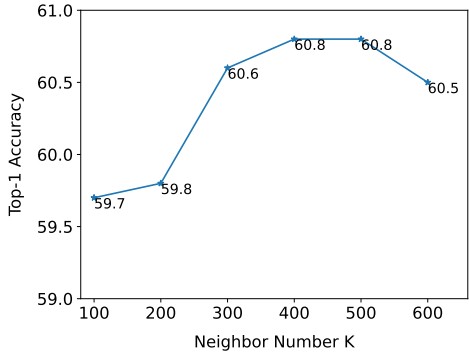

Figure 11: Impact of neighbor number $K$.

### F.8.3 PROXIMITY GRAPH UPDATE INTERVAL $t$

Proximity graph will be updated per $t$ training iterations, and to analyze the impact of $t$, we vary $t$ in the range of {50,100,200,400} and summarize the results in Table 15. It can be observed that update intervals that are too short ($t = 50$) or too long ($t = 400$) will degrade the performance. The possible reason is that sampling on a proximity graph that is frequently updated results in unstable learning of the model. Besides, the distribution of instances in the embedding space will change during the training process, resulting in a shift in hard negatives. As a result, after a few iterations, the lazy-updated graph cannot adequately capture the similarity relationship.

Table 15: Performance comparison with different update interval $t$ on CIFAR10 and CIFAR100.

| Update Interval $t$ | 50 | 100 | 200 | 400 |
|---|---|---|---|---|
| CIFAR10 | 92.29 | **92.54** | 92.34 | 92.26 |
| CIFAR100 | 68.37 | **68.68** | 67.83 | 68.59 |

### F.9 TRAINING CURVE

We plot the training curves on STL10 and ImageNet-100 respectively. As shown in Figure 12, on STL10 dataset, ProSampler takes only about 600 epochs to achieve the similar performance as the original SimCLR, which takes 1000 epochs. A similar phenomenon can be seen on ImageNet-100. All these results manifest that ProSampler can bring model better and faster learning.

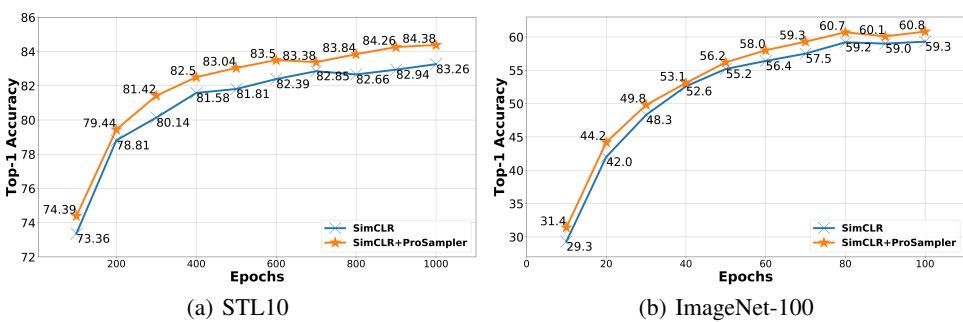

(a) STL10          (b) ImageNet-100

Figure 12: Training curves for image classification task on STL10 and ImageNet-100.

## F.10 CASE STUDY

To give an intuitive impression of the mini-batch sampled by ProSampler, we show some real cases of the negatives sampled by ProSampler and Uniform Sampler in Figure 13. For a given anchor (a cat or a dog), we apply ProSampler and Uniform Sampler to draw a mini-batch of images, and pick the images with the highest inner product with anchor. Obviously, compared with Uniform Sampler, the images sampled by ProSampler are more semantically relevant to the anchor in terms of texture, background or appearance.

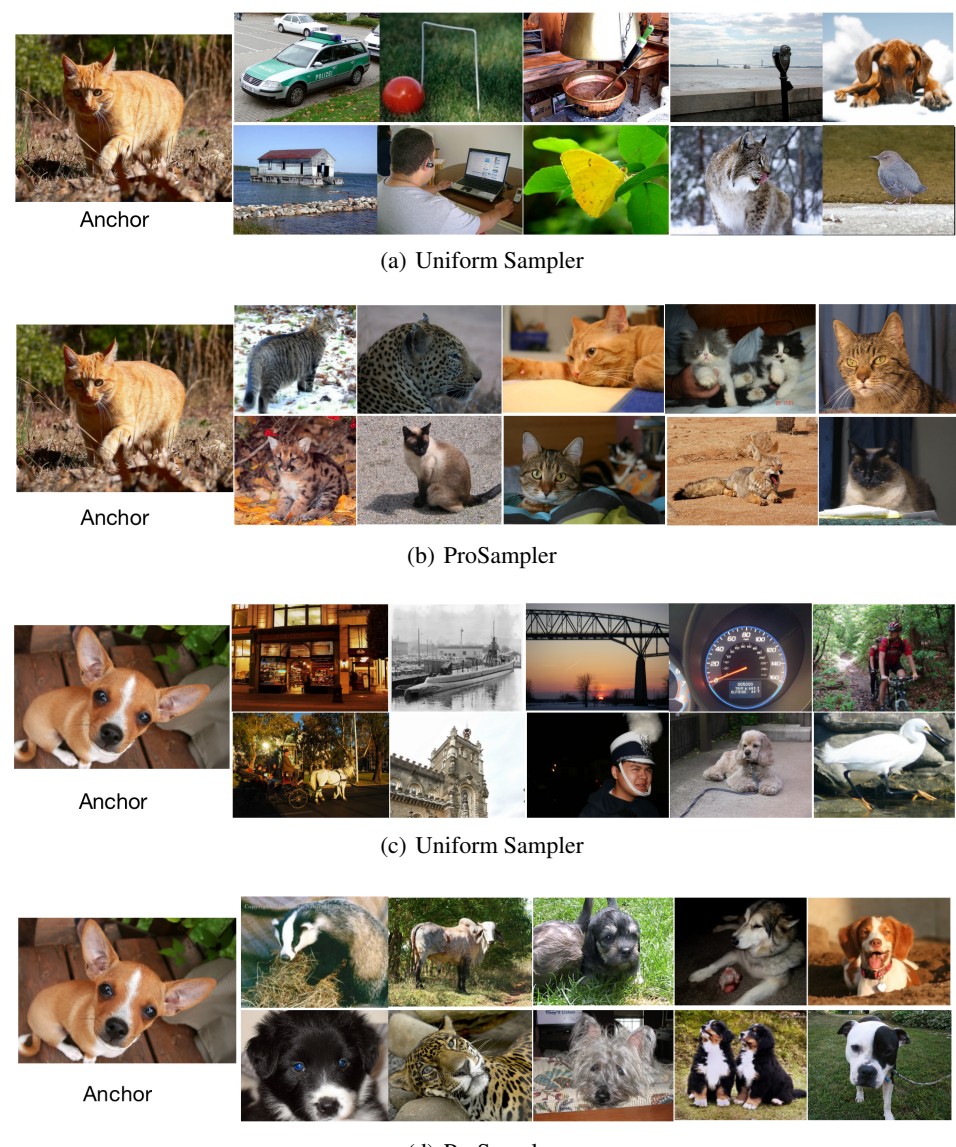

(a) Uniform Sampler

(b) ProSampler

(c) Uniform Sampler

(d) ProSampler

Figure 13: Case study of the negatives sampled by ProSampler and Uniform Sampler based on the encoder trained for 100 epochs on ImageNet. Given an anchor image (Cat or Dog), **(a,c)** select 10 images with the highest similarity from a mini-batch sampled by ProSampler, and **(b,d)** randomly select 10 images from a mini-batch sampled by Uniform Sampler.

