# OpenReview forum: "ProSampler: Improving Contrastive Learning by Better Mini-batch Sampling"
_ICLR.cc/2023/Conference — Submitted to ICLR 2023_

### Official Review · Reviewer_uJHq · 2022-10-15

**Confidence:** 5
**Correctness:** 4
**Technical Novelty And Significance:** 3
**Empirical Novelty And Significance:** 3
**Recommendation:** 8

**Clarity, Quality, Novelty And Reproducibility:**

The novelty lies in the problem framing: there are many hard negative sampling methods for contrastive leanring, but many either 1) work by mixing, reweighing etc. samples from a uniformly sampled mini-batch, or 2) add some additional mechanism such as a memory bank. The proposed method is well distinguished from these approaches, by keeping the SimCLR-style in-batch negative sampling, but more intelligently sampling the batch itself to ensure harder samples.

Approach comes with adequate implementation details. I **strongly encourage** the authors to release their code, which they do not currently state that they plan to do.

Clarity is also good, but can always be improved. Take some care to make sure all details are explained sequentially and clearly. For instance one silly thing that tripped me up for a bit was the line "for each instance $v_i$ by randomly picking $M(M < N)$ neighbor candidates." I wasn't clear how these $M$ samples were being picked - was it uniformly? Or using some other sampling algorithm? After searching and finding no further description of how this sampling is done I concluded it was uniformly. But you can save the reader this time by being precise and stating that it is uniform.

**Strength And Weaknesses:**

**Strengths**

- A good, broad set of experiments show that the approach, called ProSampler, lead to better downstream performance on many tasks (images, text, graphs). They also show that ProSampler can be successfully combined with existing in-batch hard negative sampling methods such as that of Robinson et al.

- Methodological studies show that ProSampler behaves as expected in a number of ways: for instance it samples negatives that are harder (higher cosine similarity) than uniform sampling, but less similar than the $k$ nearest neighbors (hardest negatives possible).

- My personal main strength is the proposal to use graphs based on input similarities for batch sampling. Many other alternative such methods could be tried too.

**Weaknesses**

- the two additional hyperparameters will probably be a bother to users. At least they appear to be reasonably robust, so decent default choices can be used.

- the paper probably could have been made more enlightening by focusing more on proposing a more general underlying principle (graphs build from input similarities) and talking more about how this graph should be built. At the moment, the proposed method is simply given, and it is very believable. But more work is left to the reader than is necessary terms of




**Summary Of The Paper:**

This paper proposed a batch sampling method aimed at generating harder negative samples for contrastive learning methods that use other samples in a batch as negatives, for instance SimCLR.

Brief description of the method: the approach builds a graph whose nodes are training datapoints, and each node is connected to the $k$ nearest neighbors (distances measured using stale embeddings) of a uniformly subsampled graph (the subsampling stochastically ensures there is some connections between less similar samples). A batch of size $B$ is generated by selecting a random starting node, and running a random walk until $B$ distinct nodes have been visited. In order to ensure the sampling process doesn't drift too far from the starting point, each random walk step has a fixed probability of returning to the starting node.



**Summary Of The Review:**

Congratulations on a nice paper. The approach is:

- well motivated: global methods for generating hard negatives are likely to work better that in-batch methods.
- sensibly executed: the proposed algorithm, based on constructing a similarity graph and running a random walk, is a natural approach building on core graph algorithms. There will be many many other possible implementations people could test too. For instance, simply define the weight $w_{ij}$ of edge $(i,j)$ to be the similarity score $e_i \cdot e_j$ between the embeddings of points $i$ and $j$. Then run a random walk where at. node i, the next node is sampled from the distribution Softmax($w_{ij}$) (softmax over all $j$'s). Probably keeping the chance of returning to the starting node at each step. My point here is that the proposed ProSampler is just a single example of many similar algorithms based on random walks on similarity graphs, many of whom will probably work well too. This is good in my books since it means that the paper hasn't just proposed a single fiddly algorithm that they managed to get to work, but that there is a core underlying idea, which is robust to many different possible implementations.
- thoroughly evaluated: downstream performance improves. on lots of tasks across three modalities. Visualizations (e.g., Fig 12) and comparisons of negative similarities (e.g., Fig 5) give good sanity checks that the method is working as hoped.


In all this is a solid piece of work and I can't see any major flaw that could justify rejection.

---

> ### Author Response · Authors · 2022-11-18
> **Reply to Reviewer uJHq**
>
> We sincerely thank you for your valuable comments on our paper. We will explain your concerns point by point.
>
>
> ```
> Q1: The two additional hyperparameters will probably be a bother to users. At least they appear to be reasonably robust, so decent default choices can be used.
> ```
> A1: These two hyperparameters are critical to the performance of ProSampler. To better understand their impact, we conduct some experiments to analyze and provide an empirical criterion for selecting them in Section 4.3.
>
>
> ```
> Q2: The paper probably could have been made more enlightening by focusing more on proposing a more general underlying principle (graphs build from input similarities) and talking more about how this graph should be built. At the moment, the proposed method is simply given, and it is very believable. But more work is left to the reader than is necessary terms of ?
> ```
> A2: Thanks for your suggestion! Indeed, how to build the graph used for conducting mini-batch sampling is an interesting question. And it is worthwhile to explore ways to balance the efficiency and effectiveness of graph construction. We experiment with several methods of building proximity graphs, and our experience shows that both the **diversity** and **similarity** of neighbors are crucial factors to the performance.
> This also serves as a motivation for the current Proximity graph construction strategy which balances diversity and similarity by randomly selecting neighbor candidates. We will provide more intuition and insights for graph construction in our following manuscript.
>
>
> ```
> Q3: Approach comes with adequate implementation details. I strongly encourage the authors to release their code, which they do not currently state that they plan to do.
> ```
> A3: Our source code is provided in Supplementary Material. Unfortunately, our submission is one of a few in which the supplementary material might have been corrupted because of a technical issue with OpenReview. We resubmitted the source code immediately after receiving the email from OpenReview.
>
> ```
> Q4: Clarity is also good, but can always be improved.
> ```
> A4: Thanks for your suggestion! We have revised our submission accordingly.

---

> ### Author Response · Authors · 2022-12-05
> **We're looking forward to your reply!**
>
> Dear Reviewer uJHq,
>
> Thank you very much again for your valuable reviews! We hope our previous response addresses your questions. If possible, we would be deeply grateful if you could provide your thoughts and suggestions on our paper. Your kind reviews are crucial to make ProSampler a better work and we would greatly appreciate your participation in the rebuttal stage.
>
> Thank you again for your encouraging comments!

---

### Official Review · Reviewer_nXV4 · 2022-10-25

**Confidence:** 4
**Correctness:** 3
**Technical Novelty And Significance:** 3
**Empirical Novelty And Significance:** 3
**Recommendation:** 6

**Clarity, Quality, Novelty And Reproducibility:**

The paper is clear, well written and easy to read. It introduces a novelty in terms of creating more informative and effective mini-bacthes for training contrastive learning methods.

**Strength And Weaknesses:**

**Strength**
- creating more informative mini-batches.
- selecting hard negatives examples "globally".
- time complexity and time cost of mini-batch sampling and proximity graph construction (*)


**Weaknesses**

- what is the space complexity? can a graph representing imagenet-20K be kept in memory?

- "Sampling on the proximity graph can better exploit the hard negatives "globally" by bridging in similar instances from the entire dataset."

1. **Proximity Graph Construction**: it is not clear if the construction of the graph is based on all the examples of the dataset or if there is a random selection of the examples.

2. Looks like "that the candidate set for each instance is formed by randomly picking a number of neighbor candidates"
  - if the candidates are drawn at random how is it possible to say that the method can "exploit the hard negatives globally"?
  - if the neighbors are given, it means that there's an additional cost about knowing which neighbors of a certain node are? how are neighbors found?





**Summary Of The Paper:**

The authors propose a new sampling strategy for the construction of batches containing hard-negatives (avoiding false negatives in the batch) for contrastive learning methods. According to previous studies, it is legitimate to assume that hard negative pairs contribute the most in training the network. The method is based on a proximity graph and the sampling is performed by short random walks (-with restart). Results show improvements over a set of different tasks and datasets.

**Summary Of The Review:**

The authors propose a new solution to the problem of selecting hard-negatives from a dataset without including high percentages of false negatives (as it happens in the KNN-sampler). The solution seems to be not expensive in computational terms, some details are missing that could clarify any doubts about the cost required for the construction of the sampler (graph-based) and memory required.

---

> ### Author Response · Authors · 2022-11-18
> **Reply to Reviewer nXV4**
>
> We sincerely thank you for your valuable comments on our paper. We will explain your concerns point by point.
>
> ```
> Q1: what is the space complexity? can a graph representing imagenet-20K be kept in memory?
> ```
> A1: The space cost of ProSampler mainly comes from two parts: graph construction and graph storage.
>
>  + Proximity graph construction requires all the embeddings generated by the model, so the space complexity is $O(Nd)$ where $N$ is the number of instances and $d$ is the embedding size.
>
> + As for graph storage, actually, how to organize a large-scale graph for machine learning has been well studied in many related works\[1,2,3\]. A graph can be organized as a set of edge indexes or a sparse adjacent matrix, and both of them can be used efficiently. In our paper, proximity graph is saved as an array, with a list of neighbor indexes in each row. It has $N\times K$ elements where $K$ is the number of neighbors. So the space complexity is $O(NK)$.
>
> In conclusion, the total space complexity of ProSampler is **$O(Nd+NK)$**. Assume that the value with int type has 4 Bytes and the number of neighbors is 500. ImageNet-21K contains 14 million images, so we require $14,000,000 * 500 * 4 / 1024^3$ = 26G memory to store the graph. We have added the space complexity analysis in the updated manuscript.
>
> ```
> Q2: It is not clear if the construction of the graph is based on all the examples of the dataset or if there is a random selection of the examples? If the candidates are drawn at random how is it possible to say that the method can "exploit the hard negatives globally"?
> ```
>
> A2: As we mentioned in Section 3.3, the proximity graph is constructed by first randomly picking $M$ neighbor candidates and then retrieving the $K$ nearest ones from the candidate set by the inner product. Although neighbor candidates are randomly drawn, the similarity between a node pair in the graph can be theoretically guaranteed, and $M$ can adjust the similarity. Specifically, a higher $M$ indicates a greater probability that two adjacent nodes are similar, and the proximity graph will be more similar to the kNN graph. Furthermore, the constructed proximity graph is **globally connected**, allowing the graph sampling method to explore hard negatives globally.
>
>
> ```
> Q3: If the neighbors are given, it means that there's an additional cost about knowing which neighbors of a certain node are? how are neighbors found?
> ```
> A3: As we mentioned in A1, how to conduct machine learning on large-scale graphs has been studied in many related works\[1,2,3\]. In our implementation, we follow these previous strategies and save the proximity graph as an **array**, with each row containing a list of neighbor indexes. The space complexity of proximity graph storage is **$O(NK)$** where $N$ is the number of instances and $K$ is the number of neighbors. Since $K$ is typically much lower than $N$, it is practically efficient. We can easily refer to a node's index number whenever we want to access its neighbors. More efficiency analysis can be found in Appendix F.5.
>
>
>
> [1] Tang, Jian, et al. "Line: Large-scale information network embedding." Proceedings of the 24th international conference on world wide web. 2015.
> [2] Grover, Aditya, and Jure Leskovec. "node2vec: Scalable feature learning for networks." Proceedings of the 22nd ACM SIGKDD international conference on Knowledge discovery and data mining. 2016.
> [3] Fey, Matthias, and Jan Eric Lenssen. "Fast graph representation learning with PyTorch Geometric." arXiv preprint arXiv:1903.02428 (2019).

---

### Official Review · Reviewer_4epo · 2022-10-26

**Confidence:** 4
**Correctness:** 3
**Technical Novelty And Significance:** 2
**Empirical Novelty And Significance:** 3
**Recommendation:** 3

**Clarity, Quality, Novelty And Reproducibility:**

- The paper is quite well-written and well-organized. The motivation is clear and convincing.

- The novelty, however, is limited for me.

- The source code is provided in supplementary.

**Strength And Weaknesses:**

### Strength
- The paper is well-organized and easy to follow. The motivation is clarified and the sampling of hard negative pairs is a promising topic.
- The experiments are sufficient for me.

### Weakness
My main concern is the limitation of the novelty of the sampling strategy. The proximity graph, the core of the proposed sampling, is constructed through two steps:
- Randomly (Uniformly) select multiple points,
- Choose the k nearest ones.
It seems like the method to introduce randomness to KNN. In other words, it seems like a combination of the two existing strategies. The subsequent random walk step is another step to introduce further stochasticity.

Although some theoretical analyses are provided, they are not attractive to me.

**Summary Of The Paper:**

This paper focuses on how to sample hard negative pairs in every batch, which is an important topic in contrastive learning. The paper motivates by the defects of uniform and KNN strategies, and then the authors propose to construct a proximity graph and employ the random walk to generate each batch. According to the construction of the proximity graph (selecting samples randomly and then choosing the k nearest ones), the sampling method can be regarded as a combination of uniform sampling and KNN.

**Summary Of The Review:**

The paper is well-written and I admit that the topic is meaningful. However, I concern whether the novelty of the sampling is significant enough for ICLR. I would like to update my score during the discussion period.

---

> ### Author Response · Authors · 2022-11-18
> **Reply to Reviewer 4epo**
>
>
> Thanks for your feedback on our work. We will address your main concern.
>
> ```
> Q1: My main concern is the limitation of the novelty of the sampling strategy. The proximity graph, the core of the proposed sampling, is constructed through two steps: 1) Randomly (Uniformly) select multiple points; 2) Choose the k nearest ones. It seems like the method to introduce randomness to KNN. In other words, it seems like a combination of the two existing strategies. The subsequent random walk step is another step to introduce further stochasticity. Although some theoretical analyses are provided, they are not attractive to me.
> ```
> A1: Overall, the goal of ProSampler is to **globally** sample hard negatives, which is different from previous methods that **locally** sample hard negatives. To achieve this, we propose to collect a batch that contains a significant number of hard-to-distinguish instance pairs to guarantee hard negatives within a batch.
>
> As **Reviewer uJHq** concludes, "The proposed method is well distinguished from these approaches, by keeping the SimCLR-style in-batch negative sampling, but more intelligently sampling the batch itself to ensure harder samples."
>
> Here, we highlight our contributions from three folds:
>
> 1) ProSampler is the **first** method on **global** hard negative sampling for in-batch contrastive learning, which focuses on collecting more hard-to-distinguish pairs in the each sampled batch.
>
> 2) The proposed framework is very **general** and can be easily plugged into **any** in-batch contrastive learning methods with three different mainstream modalities (image, text, and graph). Besides, it is efficient and can handle large-scale datasets, e.g., ImageNet;
>
> 3) We establish a **theoretical connection** between the hardness of sampled mini-batch and the behavior of ProSampler, demonstrating how the candidate set size $M$ can be used to control the similarity between the central node and its neighbors, which is essential for balancing the hardness and false negatives.
>
> Besides the above contributions, let's take a closer look at ProSampler:
>
>
> + Global hard negative sampling is an effictive method to significantly improve the model[1,2], but has been neglected in in-batch contrastive learning so far. Previous hard negative sampling methods, like mixing and reweighting, **locally** select hard negatives within a uniformly sampled mini-batch. However, such strategies cannot exploit hard negatives sufficiently from the whole dataset. We focus on **globally** sampling hard negatives to improve the downstream performance of in-batch contrastive learning.
>
> + As we discuss in Section 3.1, the **key challenge** of global hard negative sampling for in-batch contrastive learning is how to guarantee similarity among the instances in the sampled mini-batch. **kNN Sampler** is a straightforward method to achieve it by retrieving the nearest neighbors to construct a mini-batch, but it performs poorly due to the false negative issue (See Figure 4). To obtain a desired batch that contains considerable hard-to-distinguish pairs, we propose **ProSampler** that performs mini-batch sampling as graph sampling on the constructed proximity graph, allowing it to globally explore the hard negatives and flexibly control the hardness. As shown in the following table, ProSampler can significantly outperform the kNN Sampler with an absolute improvement of 2.38% and 6.30%.
>
> |Method | CIFAR10 | CIFAR100|
> |----|----|----|
> |kNN Sampler | 90.16 | 62.30 |
> |ProSampler | 92.54 | 68.68 |
>
>
> + Technically, ProSampler leverages a proximity graph to capture **similarity** relationships among instances, which guarantees instances that appear to be close to each other in the graph (See Proposition 1). ProSampler performs the mini-batch sampling as a **walking** (i.e. graph sampling) to obtain a desired batch. Random Walk with Restart (RWR) ensures the graph sampling process within a local cluster rather than drifting too far from the start instance (See Proposition 2). The **hardness** of a sampled batch can be modulated by $M$ and $\alpha$ (See Figure 3). We believe that a simple but effective method is what we need in practice when applying global hard negative sampling to in-batch contrastive learning.
>
>
> [1] Xiong, Lee, et al. "Approximate nearest neighbor negative contrastive learning for dense text retrieval." arXiv preprint arXiv:2007.00808 (2020).
> [2] Wang, Wenhui, et al. "Vlmo: Unified vision-language pre-training with mixture-of-modality-experts." arXiv preprint arXiv:2111.02358 (2021).

---

> > ### Comment · Reviewer_4epo · 2022-12-06
> > **Many thanks**
> >
> > I greatly thank the authors and area chair for their great efforts.
> > I agree that the global sampling is somewhat new. My major concern is the limited novelty of designing a method incorporating the cores of kNN sampler and random sampler. I do admit that the paper provides some new ideas and it has some merits. But I'm not sure whether the novelty is enough for ICLR. I also found that Reviewer FNA5 raised the same concern. My initial score should be regarded as ~4.5. Although the authors kindly offer the response, my concern is not completely addressed after reading it.

---

> > > ### Author Response · Authors · 2022-12-07
> > > **A further explanation of the novelty and contribution**
> > >
> > > Thanks for your response. We regret there might be a misunderstanding here. In order to further clarify the novelty and contribution of ProSampler, we would like to explain the following:
> > >
> > > 1. Our method goes beyond simply incorporating kNN graphs and random walk, as we propose to perform random walk with restart on the constructed proximity graph in order to effectively balance the hardness and false negatives. Furthermore, our paper has already demonstrated the poor performance of using kNN graphs.
> > > 2. We want to emphasize that a simple and effective mini-batch sampling strategy is necessary for practical applications of contrastive learning on large-scale datasets. This philosophy is also followed by many previous works, such as DCL[1] and HCL[2], which simply modify the weighting function in the loss function.
> > > 3. Reviewer FNA5 may have misunderstood the unique aspects of ProSampler and its relationship to other methods that use graph structures in contrastive learning, such as WCL[3]. In our response, we have clearly demonstrated the differences in terms of objective and methodology. Furthermore, we have verified that by showing the improved performance of WCL equipped with ProSampler.
> > >
> > > [1] Chuang, Ching-Yao, et al. "Debiased contrastive learning." Advances in neural information processing systems 33 (2020): 8765-8775.
> > > [2] Robinson, Joshua, et al. "Contrastive learning with hard negative samples." arXiv preprint arXiv:2010.04592 (2020).
> > > [3] Zheng, Mingkai, et al. "Weakly supervised contrastive learning." Proceedings of the IEEE/CVF International Conference on Computer Vision. 2021.

---

> ### Author Response · Authors · 2022-12-05
> **We're looking forward to your reply!**
>
> Dear Reviewer 4epo,
>
> Thank you very much again for your valuable reviews! We hope our previous response addresses your questions. If possible, we would be deeply grateful if you could provide your thoughts and suggestions on our paper. Your kind reviews are crucial to make ProSampler a better work and we would greatly appreciate your participation in the rebuttal stage.
>
> Thank you again for your encouraging comments!

---

### Official Review · Reviewer_FNA5 · 2022-10-27

**Confidence:** 5
**Clarity, Quality, Novelty And Reproducibility:** Please refer to my detailed comments …
**Correctness:** 3
**Technical Novelty And Significance:** 2
**Empirical Novelty And Significance:** 2
**Recommendation:** 3

**Strength And Weaknesses:**

Strengths:

(1) The proposed method is simple and effective.

(2) The authors provide a detailed theoretical analysis.

(3) The authors conduct extensive experiments for validation, including several large datasets and different modalities.

Weaknesses:

(1) The proposed method is very simple, which is similar to searching hard negative samples from a top knn graph constructed by different similarity computation strategies. The overall contribution is incremental from this point. I wonder why the authors pick M neighbors by distance and select K nearest ones by inner product operation. And why do not you directly select the K nearest neighbors?

(2) To a certain extent, the construction of a mini-batch with hard negative samples is similar to filtering out the positive samples from the original mini-batch. There are already several methods [1,2] that adopt affinity graphs to select positive pairs from the original mini-batch and achieve much better results on contrastive feature learning and clustering tasks. In this case, the novelty of this paper is unsatisfying.

(3) The main motivation of this paper lies in the influence of batch size on existing contrastive learning methods. According to MoCo v2 and MoCo V2+, the influence of batch size is marginal. I wonder whether the proposed sampling strategy still works for such a situation and please give more explanations.

(4) According to Table 1, the authors only present results of 100 and 400 epochs. I wonder whether the final results can be improved by the ProSampler, such as the results after 800 or 1,000 epochs. Besides, compared with SimCLR, SwAV, and BYOL, the improvement on ImageNet by [1] is much more significant than the proposed method. Please compare the results with [1] in detail.

(5) According to Table 9, the improvement is marginal on these small datasets.


[1] Weakly Supervised Contrastive Learning, ICCV 2021
[2] Graph Contrastive Clustering, ICCV 2021



**Summary Of The Paper:**

This paper presents a negative sampling strategy for contrastive learning, which many contrastive learning-related frameworks can incorporate to improve performance. Instead of the uniform sampler and kNN sampler, the proposed proximity graph could well capture the similarity relationships among instances. Random walks among the proximity graph can ﬂexibly explore the negatives. Experiments on several datasets demonstrate the superiority.

**Summary Of The Review:**

The proposed method is quite simple and is highly related to existing graph-based positive sample selection methods. The overall novelty and contribution are incremental. Besides, the experimental improvement is also marginal.

---

> ### Author Response · Authors · 2022-11-19
> **Reply to Reviewer FNA5  (Part I)**
>
> We sincerely thank you for the insightful comments!
>
> Briefly, this work focuses on **globally** sampling hard negatives in contrastive learning, instead of **locally** sampling hard negatives by mixing and reweighting that have previously been examined in the literature. The key challenge is how to globally collect hard-to-distinguish pairs in the sampled mini-batch. To address this, ProSampler is proposed to globally explore the hard negatives and flexibly control the hardness.
>
> We very much appreciate your acknowledge of the goal of this work---''ProSampler is a simple and effective method that can be incorporated into many contrastive learning-related frameworks to improve performance on different modalities, which also provides a detailed theoretical analysis.'' Next, we address your questions point by point.
>
> ```
> Q1: The proposed method is very simple, which is similar to searching hard negative samples from a top knn graph constructed by different similarity computation strategies. The overall contribution is incremental from this point. I wonder why the authors pick M neighbors by distance and select K nearest ones by inner product operation. And why do not you directly select the K nearest neighbors?
> ```
> A1: ProSampler is **different from** searching hard negative samples from a kNN graph:
>
> + Conceptually, the idea of this work is to **globally** collect more hard-to-distinguish pairs in the mini-batch (Cf. Section 3.2). **kNN Sampler** could achieve this by retrieving the nearest neighbors to construct a mini-batch, but by design it faces the challenge of false negatives. Motivated by this, what ProSampler does is to balance the hardness and false negatives by introducing the candidate set size $M$. In addition, we theoretically show that $M$ is able to control the similarity between a node pair in the proximity graph. Specifically, a larger $M$ indicates a higher probability that the node pair is close in the embedding space and vice versa.
>
> |Graph Construction Method | CIFAR10 | CIFAR100|
> |----|----|----|
> |kNN graph | 90.47 | 62.67 |
> |proximity graph | 92.54 | 68.68 |
>
> + Empirically, we perform an ablation study by replacing the proximity graph with the kNN graph that directly selects $k$ neighbors with the highest scores for each instance from the whole dataset. The results are shown in the table above, suggesting that ProSampler with the proximity graph outperforms kNN graph significantly. The reason is that sampling on kNN graph can retrieve so many false negatives during the training process. We plot the percentage of false negatives in the mini-batch during training (https://imgur.com/a/ArPSww4), which shows that kNN graph introduces more than **22%** false negatives in a batch while the proximity graph brings about 13% on CIFAR10. More details can be found in Appendix F.7.
>
> + In summary, ProSampler is the very first attempt to globally and provably collect more hard-to-distinguish pairs in the sampled batch for in-batch contrastive learning, which consistently improves the performance of different contrastive learning frameworks on three modalities.
>
> ```
> Q2: To a certain extent, the construction of a mini-batch with hard negative samples is similar to filtering out the positive samples from the original mini-batch. There are already several methods [1,2] that adopt affinity graphs to select positive pairs from the original mini-batch and achieve much better results on contrastive feature learning and clustering tasks. In this case, the novelty of this paper is unsatisfying.
> ```
> A2: Despite the fact that ProSampler and these approaches\[1,2\] all use graph structure, it is both conceptually and technically different from them:
> 1) Motivation:
>     + Instead of **locally** generating hard negatives by mixing and reweighting, ProSampler focuses on **globally** sampling hard negatives from the globally connected proximity graph.
>
>     + Except using augmentations from the **same** image as positive pairs, WCL[1] and GCC[2] aim to explore additional **positive pairs** from **different** images.
>
> 2) Methodology:
>     + ProSampler gathers a desired batch with more hard-to-distinguish pairs from the proximity graph, which is built from the entire dataset **globally**, guaranteeing a sampled batch with global difficult-to-distinguish pairs.
>
>     + WCL\[1\] and GCC\[2\] instead randomly sample a mini-batch first, build the affinity graph based on the current mini-batch **locally**, and utilize the affinity graph to construct **positive pairs** for each instance in the current batch.
>
> By design, ProSampler aims to be a general plugin for in-batch contrastive learning, and thus it can be directly applied to different kinds of contrastive learning methods. That said, ProSampler can be directly used to improve WCL[1]. The performance of WCL (by using the default setting) and WCL with ProSampler on ImageNet for 100 epochs  is reported in A4.

---

> > ### Author Response · Authors · 2022-11-19
> > **Reply to Reviewer FNA5 (Part II)**
> >
> > ```
> > Q3: The main motivation of this paper lies in the influence of batch size on existing contrastive learning methods. According to MoCo v2 and MoCo V2+, the influence of batch size is marginal. I wonder whether the proposed sampling strategy still works for such a situation and please give more explanations.
> > ```
> > A3: Most likely, there might be a misunderstanding here. As shown in the third paragraph of Introduction, the motivation of ProSampler is that **hard negative** is critical to the performance of contrastive learning while its focus is not on **batchsize**. Both MoCo v2 and MoCo v2+ are memory-based contrastive learning methods that maintain a fixed-size memory bank to store negatives. But the proposed ProSampler works for in-batch contrastive learning methods, e.g., SimCLR and MoCo v3, which apply the in-batch negative sharing strategy. The results on MoCo v3 are shown in Table 1.
> >
> > ```
> > Q4: According to Table 1, the authors only present results of 100 and 400 epochs. I wonder whether the final results can be improved by the ProSampler, such as the results after 800 or 1,000 epochs. Besides, compared with SimCLR, SwAV, and BYOL, the improvement on ImageNet by [1] is much more significant than the proposed method. Please compare the results with [1] in detail.
> > ```
> >
> > A4: Our experimental results in Table 1,2,3 provide evidence that the proposed ProSampler can consistently improve the performance of the backbone contrastive learning framework in different modalities. Following your suggestion, we report the results for **800 epochs** in the table below, which shows that ProSampler consistently makes advancements.
> >
> > |Method| 100 ep| 400 ep | 800 ep |
> > |----|----|----|----|
> > |SimCLR|  64.0   | 68.1 | 68.7 |
> > |w/ ProSampler|  64.7  | 68.6 | 69.2 |
> > |MoCo v3|  68.9  | 73.3 | 73.8 |
> > |w/ ProSampler|  69.5  | 73.7 | 74.2 |
> >
> >
> > As discussed above, the objectives of ProSampler and WCL[1] are different. ProSampler focuses on collecting more hard-to-distinguish pairs in the sampled batch for in-batch contrastive learning, while WCL[1] adopts affinity graphs to select positive pairs from the current mini-batch. The proposed ProSampler can also be plugged into WCL, and experimental results are reported below.
> >
> > |Method|ImageNet|
> > |----|----|
> > |WCL|  68.1   |
> > |WCL w/ ProSampler|  68.4  |
> >
> >
> > ```
> > Q5:  According to Table 9, the improvement is marginal on these small datasets.
> > ```
> > A5: We conduct experiments on both large and small datasets for three modalities, and the results in the large datasets show that ProSampler can consistently improve in-batch contrastive learning. For example, a relative improvement of 2.5% for SimCLR on ImageNet-100, 1.4% for SimCSE on the standard STS task, and 1.2% for GraphCL on the COLLAB dataset.
> >
> > Intuitively, generating a hard mini-batch has relatively less impacts on a small dataset than it does on a bigger dataset, as ProSampler focused on global hard negatives. We provide our conjecture below:
> >
> > + The false negative (FN) issue is more pronounced on these small datasets since it is easier to pick samples of the same label on a dataset with fewer categories, which can be demonstrated in Figure 4 and Figure 8. We can find that the CIFAR10 (10 classes) has a significantly greater percentage of FNs than CIFAR100 (100 classes).
> >
> > + The backbone in-batch contrastive learning methods already achieve relatively high performance on these small datasets, e.g., accuracy of 92.13% on CIFAR10, so the room to further improve them is relatively limited.
> >
> >
> > [1] Zheng, Mingkai, et al. "Weakly supervised contrastive learning." Proceedings of the IEEE/CVF International Conference on Computer Vision. 2021.
> > [2] Zhong, Huasong, et al. "Graph contrastive clustering." Proceedings of the IEEE/CVF International Conference on Computer Vision. 2021.

---

> > ### Comment · Reviewer_FNA5 · 2022-12-12
> > **Updated comments**
> >
> > Thanks for the detailed response and additional results. After reading the response and other reviewers' comments carefully, I still think the novelty of global sampling is good enough for this top conference, especially given the existence of local sampling methods of WCL and GCC. I recognize the difference between them, which is not significant from my point of view. Besides, the improvment is still unsatisfying. Specifically, the improvment over WCL is only 0.3%. And the improvments over SimCLR and MoCo v3 are less than 0.5%. Therefore, I still think this paper is not ready for publication.

---

> > > ### Author Response · Authors · 2022-12-12
> > > **Our method is not only for Computer Vision**
> > >
> > > We are happy to see that your concern about the novelty of our work has been addressed. In terms of performance, we would like to highlight that ProSampler is a general framework that can be applied to three different modalities, not just images. We have seen significant improvements in the SimCSE and GraphCL modalities, with relative improvements of 1.4% and 1.2%, respectively. Such generality sets ProSampler apart from previous work and offers a distinct advantage.

---

> ### Author Response · Authors · 2022-12-05
> **We're looking forward to your reply!**
>
> Dear Reviewer  FNA5,
>
> Thank you very much again for your valuable reviews! We hope our previous response addresses your questions. If possible, we would be deeply grateful if you could provide your thoughts and suggestions on our paper. Your kind reviews are crucial to make ProSampler a better work and we would greatly appreciate your participation in the rebuttal stage.
>
> Thank you again for your encouraging comments!

---

### Author Response · Authors · 2022-11-18
**General Response**

We appreciate all reviewers' time and efforts in reviewing our paper. We have made some refinement on the manuscript based on all the reviewers' suggestion. Here is a list of the updates:

Section 2-Related Works: we add the discussion of WCL[1].

Section 3-Method: We add the space complexity analysis of ProSampler.

Section 4-Experiment: We present the performance of ProSampler on ImageNet over 800 training epochs in Table 1.

Appendix F.7: We add an ablation study to compare the performance of ProSampler with proximity graph and kNN graph.

We hope our responses can clarify all your confusion and alleviate all concerns. We thank all reviewers again. Looking forward to your reply!

[1] Zheng, Mingkai, et al. "Weakly supervised contrastive learning." Proceedings of the IEEE/CVF International Conference on Computer Vision. 2021.

---

### Decision · Program_Chairs · 2023-01-20

**Decision:**

Reject

**Justification For Why Not Higher Score:**

There is just not enough here to warrant recommendation to accept.  Two of the reviewers are positive but the issues raised by the negative reviewers did not seem to be adequately addressed.

**Justification For Why Not Lower Score:**

N/A

**Metareview: Summary, Strengths And Weaknesses:**

Thanks for your submission to ICLR.

This paper ultimately had two positive and two negative reviews, which continued to be the case after discussion.  On the positive side, reviewers noted that the method was simple, with a theoretical analysis, and solid experiments.  On the negative side, two reviewers noted that the approach was incremental and similar to existing techniques, as well as having unclear practicality.

The reviewers both commented during the discussion phase that their initial concerns were still present even after the rebuttal.  With only two of the four reviewers advocating for accepting the paper, and lingering questions remaining about the novelty and the method's incremental nature, I am going to recommend a reject.

**Summary Of Ac-Reviewer Meeting:**

I attempted to organize an AC-reviewer meeting but some of the reviewers did not respond even after repeated emails.  Thus, we were not able to hold the meeting.